

# Experimental and model assessment of PM2.5 and BC emissions and concentrations in a Brazilian city – the Curitiba case study

Lars Gidhagen[*1], Patricia Krecl[2], Admir Créso Targino[2], Gabriela Polezer[3], Ricardo H. M. Godoi[3], Francisco Castelhano[4], Erika Felix[5], Yago Alonso Cipoli[2], Francisco Malucelli[6], Alyson Wolf[7], Marcelo Alonso[8], David Segersson[1], Jorge Humberto Amorim[1], Francisco Mendonça[4]

[1] Swedish Meteorological and Hydrological Institute (SMHI), Norrköping, Sweden
[2] Federal University of Technology, Graduate Program in Environmental Engineering, Londrina, PR, Brazil
[3] Federal University of Paraná, Environmental Engineering Department, Curitiba, PR, Brazil
[4] Federal University of Paraná, Department of Geography, Curitiba, PR, Brazil
[5] Federal University of Technology, Department of Chemistry and Biology, Curitiba, PR, Brazil
[6] Institute for Research and Urban Planning of Curitiba (IPPUC), Curitiba Municipality, Brazil
[7] Curitiba Urbanization (URBS), Curitiba Municipality, Brazil
[8] Federal University of Pelotas, Faculty of Meteorology, Pelotas, RS, Brazil

*Correspondence to*: Lars Gidhagen (lars.gidhagen@smhi.se)

**Abstract.** Data on airborne fine particle emissions and concentrations in cities are valuable to traffic and air quality managers, urban planners and landscape architects, health practitioners, researchers, and ultimately to legislators and decision makers. This study aimed at determining the emissions and ambient concentrations of black carbon (BC) and fine particles (PM$_{2.5}$) in the city of Curitiba, southern Brazil. The methodology applied combined a month-long monitoring campaign that included both fixed and mobile instruments, the development of emission inventories, and the dispersion simulation from the regional down to the street scale.

The mean urban background PM$_{2.5}$ concentrations during the campaign were below 10 µg m$^{-3}$ in Curitiba city center, but two- to three-fold higher in a residential area, indicating the presence of unidentified local sources, possibly linked to wood combustion. Mean BC concentrations seemed to be more uniformly distributed over the city, with urban background levels around 2 µg m$^{-3}$, which rose to about 5 µg m$^{-3}$ in heavily trafficked street canyons. The dispersion modeling also showed high PM$_{2.5}$ and BC concentrations along the heavily transited ring road and over the industrial area southwest of Curitiba. However, the lack of in situ data over this area prevented the corroboration of the model outputs. The integrated approach used in this study can be implemented in other Brazilian cities as long as an open data policy and a close cooperation between municipal authorities and academia can be achieved.

## 1 Introduction

It is generally accepted that the fast pace of global urbanization is associated with the expansion of modern industries and non-agricultural sectors, especially in low- and middle-income countries (Chen et al., 2014). The urbanization and




industrialization have increased the concentrations of outdoor air pollution from vehicles and factories, contributing to the global burden of asthma and allergic diseases (Zhang et al., 2015). A recent global model assessment with relatively high spatial resolution (11x11 km$^2$) revealed that the human exposure to ambient $PM_{2.5}$, i.e., fine particles with an aerodynamic diameter smaller and equal to 2.5 µm, ranked fifth as mortality risk factor in 2015 with more than four million deaths (Cohen

et al., 2017). The urgent need to take better care of cities' environment has recently been manifested in the Sustainable Development Goal (SDG) 11 addressing urban planning and development. One specific SDG indicator, 11.6.2, for meeting this goal is the annual mean level of $PM_{2.5}$ weighted by population (United Nations Statistics Division, 2018), for which a global database was developed by the World Health Organization (WHO, 2016).

Ongoing research is trying to identify which components of inhalable particulate matter contribute the most to the observed

health effects. The Review of Evidence on Health Aspects of Air Pollution (REVIHAAP) assessment (WHO, 2013) could not pinpoint the constituents within the $PM_{2.5}$ matrix that are more related to specific health outcomes. However, strong research outcomes have shown that black carbon (BC) particles - associated to combustion emissions, especially from diesel engines - are a more robust indicator of health effects than solely PM2.5 (Janssen, et al., 2011). An additional aspect of BC as a metric for air pollution is its potential to act as a Short-Lived Climate Forcer (SLCF), contributing to warm up the

atmosphere (Bond et al., 2013). Hence, curbing BC emissions has the double benefit of reducing the human exposure to $PM_{2.5}$, while contributing to mitigate climate change.

Global assessments of air pollution have revealed that the highest $PM_{2.5}$ exposure levels are found in Asia, whereas lower exposure occurs in the American continent (WHO, 2016). However, there is no threshold for the $PM_{2.5}$ health effects below which the consequences for human health are negligible. On the contrary, there are indications that a reduction of 1 µg m−3

on $PM_{2.5}$ concentrations provides a more significant health benefit in a city with relatively low levels (i.e., in the range of 10-30 µg m$^{-3}$), as compared to cities with much higher mean concentrations, close or above 100 µg m$^{-3}$ (Burnett et al., 2014). Another aspect, illustrating the importance of reducing BC, is exemplified by Janssen et al. (2011), who showed that a 1 µg m$^{-3}$ decrease in $PM_{2.5}$ exposure would lead to an increase in life expectancy of 21 days per person, whereas the same reduction in BC concentration would yield an increase between 3.1 and 4.5 months.

The national ambient air quality standards (NAAQS) in Brazil were implemented in 1990 and legislate particulate matter concentrations as total suspended particles, $PM_{10}$, and black smoke (another metric for exhaust emissions). However, only 12 of the 27 federal units of Brazil have at least one air quality monitoring station (Instituto de Energia e Meio Ambiente, 2018). Black smoke is regulated with very tolerant thresholds (mean daily value of 150 µg m$^{-3}$ and mean annual value of 60 µg m$^{-3}$), but monitoring occurs in only 11.7% of all stations in the country (Instituto de Energia e Meio Ambiente, 2018).

São Paulo is the only state in Brazil where a $PM_{2.5}$ standard has been implemented, with limit values of 60 µg m$^{-3}$ and 20 µg m$^{-3}$ for the daily maximum and annual average, respectively.

With no legislation supporting the monitoring of $PM_{2.5}$ and BC, most Brazilian cities lack information on these pollutants, except for a few short-term campaigns, which have been conducted to address specific aspects of atmospheric processes (Krecl et al., 2018; Polezer et al., 2018; De Miranda et al., 2012; Targino and Krecl, 2016). The significant health and



climate benefits of reducing PM$_{2.5}$ and BC emissions should be an incentive for the Brazilian environmental agencies to assess their concentrations and spatial distribution across cities and to identify their source contributions. However, given financial and infrastructure constraints, uneven spatial distribution of stations, lack of instrument maintenance, spare parts and technical expertise, the operational monitoring of PM$_{2.5}$ and BC concentrations in Brazil is still challenging. A positive

note is that there are plans under preparation in some states (e.g., Paraná) to formulate limit values for PM$_{2.5}$ in line with what São Paulo has already implemented. In parallel, there is an urgent need to fill the knowledge gap in terms of local emissions of PM$_{2.5}$ and BC.

A set of recommendations has been proposed by the General Assembly of the World Medical Association (WMA) in 2014, including: a) monitoring and limiting the concentrations of nanosize BC particles in urban areas, b) building professional and

public awareness of the hazard of BC and the existing methods of eliminating the particles, c) developing strategies to protect people' exposure to BC in motorized transport, homes and in the general environment (World Medical Association, 2014). In line with WMA's guidance, the present work is the result of a two-year project entitled ParCur ("Particles in Curitiba"), conducted in Curitiba, the capital city of the state of Paraná. The ParCur project has formed part of a bilateral cooperation between Brazil and Sweden. In line with the SDG objectives, as well as with SLCF reduction initiatives such as

the Climate and Clean Air Coalition (CCAC), Sweden supports bilateral cooperation with specific countries within environmental protection, climate change, and sustainable development.

ParCur gathered experts from Brazil and Sweden, local stakeholders and end users. As far as we know, this is a pioneering study in South America with the integration of fixed and mobile high spatio-temporal resolution PM$_{2.5}$ and BC measurements, the development of emission inventories and the implementation of modelling tools at different spatial scales

for validating the emissions and to determine the spatial distribution of pollutant concentrations across the city.

## 2 Method

### 2.1 Study area

Curitiba is located in Southern Brazil on a plateau at approximately 900 m above sea level and at a distance of 110 km from the Atlantic Ocean. The Curitiba municipality has an estimated population of 1.9 million inhabitants. From the urbanistic

standpoint, the Serete Plan, created in 1964 and implemented in the 1970's, was the backbone of the development plan that shaped the city's current structure and morphology (Santos, 2014). This plan organized the municipality using the triad: public transportation, land use, and road system, which led to the creation of the structural axes or transport corridors, some dedicated to the Bus Rapid Transit (BRT) system which was pioneered in Curitiba. An important feature of the city´s planning is the separation of the industrial areas from the city center, located in the southwest, downwind of the dominating

northeasterly winds.

The official air quality monitoring network is managed by the Environmental Institute of Paraná (IAP) and consists of four automatic stations within the municipality of Curitiba: Boqueirão (BOQ), Cidade Industrial (CIC), Ouvidor Pardinho (PAR),



Santa Cândida (STC) and four in the industrial area of the nearby city of Araucária. An analysis of $PM_{10}$ and $NO_2$ data from the official monitoring network was performed for the three year period 2013 to 2015. The objective was to describe the general pollution levels, extreme values and seasonal variability, for a period close to the campaign performed as part of this project in 2016, i.e., for conditions with similar urban emissions.

**2.2 Study design**

A combination of measurements and dispersion modeling was used to assess and map $PM_{2.5}$ and BC concentrations. The modeling had two purposes: validating the emission inventory through comparisons with measured data at a few stations, and obtaining a spatial distribution of the $PM_{2.5}$ and BC concentrations over the city. The comparison of monitored data and model output can be used for microenvironments impacted by one dominating source to allow an in situ determination of the

emission factors, in this case for vehicles driven in the city.

The full assessment included the following components (see map in Fig. 1):

1. Analysing the $NO_2/NO_x$ and $PM_{10}$ concentrations collected between 2013 and 2015 at the four IAP official monitoring sites in Curitiba: PAR, BOQ, STC and CIC

2. Developing an emission inventory for the city and the state of Paraná.

3. Performing field campaign at two fixed sites (Fig. 1 left) aimed at:

    a) monitoring of $NO_x$, $PM_{2.5}$ and BC concentrations within a street canyon (Marechal Deodoro, hereafter MD) in the city center at two levels above ground: street (height of 5 m, hereafter 'MD street') and rooftop (height of 70 m, hereafter 'MD roof').

    b) monitoring of $PM_{2.5}$, BC, EC (elemental carbon) and OC (organic carbon) concentrations in a residential area (Sítio
Cercado, hereafter SC) located 13 km from the city center, and 750 m from the heavily trafficked road BR-376, which is part of Curitiba's ring road ('Contorno' as local designation).

4. Performing a monitoring campaign with instruments on-board bicycles to measure $PM_{2.5}$ and BC concentrations along different types of roads in the city center (see Fig. 1, right).

5. Implementing dispersion models at the regional, urban and street canyon scales to support the interpretation of the
monitored data.

6. Consolidating the street canyon data and model output to obtain real-world emission factors for $PM_{2.5}$ and BC for road transport in Curitiba.

7. Using the regional and urban modeling, together with the monitored data, to conduct a source apportionment of $PM_{2.5}$ and BC levels in Curitiba.

**2.3 Emission inventory**

The emission inventory developed for Curitiba considered $PM_{2.5}$, BC and $NO_x$ for two major economic sectors: industries and on-road transport. An attempt was made to collect data on the use of wood or coal by restaurants. However, the database



gathered was incomplete in space, impeding the inclusion of this source in the emission inventory. Neither was it possible to obtain data on the residential use of wood stoves for cooking or heating; however, municipal authorities informed that residential wood combustion should be minimal, at least in the city center.

Industrial emission values from large industrial sources were compiled from the official regional inventory that covers the
state of Paraná (IAP, 2013), while the Curitiba municipality provided data for the inner-city small scale industries. Because these industrial inventories only included $PM_{10}$ emissions, we assumed that 70% of the $PM_{10}$ emitted by the industries in and around Curitiba consisted of $PM_{2.5}$ (Erlich et al., 2007). All industrial emissions were treated as point sources with emissions coming out of a stack with characteristics given by the IAP inventory.

Traffic emissions in Paraná state, but outside the city of Curitiba, were also extracted from the official regional inventory
(IAP, 2013). Within the city, the emission calculations were split between public transport (buses) and private vehicle emissions. Public transport emissions were calculated as line sources along the bus network, as facilitated by the municipality. Information on bus size, technology (Table 1), average daily distance travelled and average fuel consumed were obtained for each bus line. Together with the bus timetables, it was possible to describe the number of buses transiting a certain road link on an hourly basis. We used a relation between the Brazilian emission legislation PROCONVE and the
European EURO classes (TransportPolicy.net, 2018a; 2018b), enabling the use of emission factors from the European HBEFA database (INFRAS, 2017), as listed in Table 2. We assumed an average bus speed of 50 km h$^{-1}$ and saturated traffic conditions. The bus types listed in Table 1 were aggregated into HBEFA bus classes with weights <15 ton, 15-18 ton, and >18 ton, respectively. Emission factors for bi-articulated buses were extrapolated using information on fuel consumption provided by the Curitiba municipality. BC emission factors were taken from EEA (EEA, 2016) as BC/$PM_{2.5}$ fractions in the
range 65-75%. The use of biofuel lowers the PM and BC emissions by 50%, according to the U.S. Department of Energy (2018).

Private vehicle emissions were calculated based on traffic volumes per road link (excluding smaller secondary roads), simulated by the VISSIM model (VISSIM, 2018) for morning and afternoon peak hours. Hourly variations of traffic volume over the day were taken from 230 speed instruments monitoring the speed of individual vehicles and aggregated to different
profiles according to the day of the week: Monday-Thursdays, Fridays, Saturdays and Sundays. The same daily profiles were used for all streets.

As for the fleet composition, a simplified assumption was made with three different compositions depending on the type of road. The following shares were adopted, as suggested by the traffic engineers at the Curitiba Municipality: for inner-city roads (as limited by the ring road): 93% cars, 5% utility vehicles and 2% trucks; for a few larger thoroughfares open for
heavy-duty vehicles: 82% cars, 8% utility and 10% trucks; and for the ring road: 59% cars, 9% utility and 32% trucks. Table 3 shows the assumptions made on size and technology, together with emission factors taken from the European Environment Agency (EEA, 2016). Utility vehicles and trucks were assumed to be diesel-fueled, while flex-fuel cars were running on gasoline as drivers' primary fuel choice due to favorable price. Stop-and-go emissions due to congestion or traffic lights were not considered.




### 2.4 Monitoring campaign

Fixed-site measurements were conducted in the winter period July 25-August 24, 2016, when pollution levels are expected to peak (see Section 3.1 for further details). Mobile monitoring was performed on 10 selected days and times (morning and evening rush hours) between August 1 and August 14, 2016. $PM_{2.5}$ monitoring was conducted with three types of

instruments: a Harvard Impactor using cut-off 2.5 microns (deployed at street and rooftop levels of the canyon site), a MicroVol low volume air sampler (model 1100, Ecotech, Australia) at SC, and three DustTrak units (model 8520, TSI, USA), one deployed at SC and two units used for mobile measurements.

The Harvard Impactor collected $PM_{2.5}$ samples on 37-mm teflon filters for gravimetric analysis with a 24-hour resolution. Daily integrated samples for gravimetric, EC and OC analysis were collected on 47-mm quartz fiber filters using the Ecotech

MicroVol. The gravimetric analyses were performed following the NIOSH Method 5000 (NIOSH, 2003), while the EC and OC analyses were conducted at Stockholm University using the NIOSH temperature protocol (Birch, 2003). The $PM_{2.5}$ output from the DustTrak instruments was calibrated with the gravimetric data from SC, yielding a correction factor of 1.92 ($R^2 = 0.77$), which was subsequently applied to correct all DustTrak outputs. This correction factor is within ranges reported by other studies conducted in urban areas, from 1.70 (McNamara et al., 2011) to 2.78 (Wallace et al., 2011).

Total BC concentrations at street and roof levels were measured with aethalometers (models AE42 and AE33, respectively, Magee Scientific, USA) operating at seven wavelengths (370, 470, 520, 590, 660, 880, and 950 nm) and with five minute time resolution. At SC and for the mobile measurements, BC concentrations were determined with microaethalometers (model AE51, AethLabs, USA) operating at the wavelength of 880 nm. These instruments use the wavelength-dependent absorption cross-section values provided by the instrument manufacturers to convert aerosol absorption coefficient into BC

mass concentrations. In this study, a site-specific absorption cross section of 18.39 $m^2$ $g^{-1}$ was determined at SC by correlating daily mean aerosol absorption coefficients with collocated EC concentrations ($R^2 = 0.96$). The BC data from the AE33 and AE42 instruments were determined using the absorption cross section provided by the manufacturer.

The passive sampling of NOx at the roof and street level of the MD site was performed using Ogawa passive samplers (Hagenbjörk-Gustafsson et al., 2010) during two periods of 14 days each. The purpose of this $NO_x$ measurement was to

support the determination of local emission factors for vehicle emitted $PM_{2.5}$ and BC.

### 2.5 Dispersion modeling

Dispersion modeling was performed at three spatial scales with three different models, all giving hourly data for the monitoring period from July 25 to August 24, 2016. The regional scale modeling was performed with the BRAMS 5.2 modeling system (Freitas et al., 2017), which includes an atmospheric chemistry transport model (CCATT) coupled on-line

with a limited-area atmospheric model. For this specific experiment, the model was configured to simulate aerosol emission, transport and its effects (dispersion option) during campaign period. The BRAMS physical parameterizations were configured with Mellor–Yamada level-2.5 turbulence scheme (Mellor and Yamada, 1982) and Joint UK Land Environment



Simulator (JULES) surface–atmosphere interaction model (Moreira et al., 2013). To shortwave and longwave radiation schemes was used RRTMG with 1200s frequency update of the radiation trend (Iacono et al., 2008). Finally the Grell and Freitas (2014) ensemble version for deep and shallow convection and the single-moment bulk microphysics parameterization from Walko et al. (1995) were used. The model was used over two domains: one covering large parts of South America with

spatial resolution of 50x50 km$^2$ (G1, Fig. 2, left) and nested down to a 10x10 km$^2$ resolution over the state of Paraná (G2, Fig. 2, left) with anthropogenic emissions taken from a South American inventory (Alonso et al., 2010). The emissions for the state of Paraná were updated with the vehicular and industrial inventory described earlier (IAP, 2013), except for BC and OC that used information from EDGAR-HTAP (Joint Research Centre, 2018). Biomass burning sources were taken from the model 3BEM (Longo et al., 2009) and biogenic emissions of gases from MEGAN (Guenther et al., 2012). These emission

fields were generated by the preprocessor PREP-CHEM-SRC (Freitas et al., 2011).

For the urban scale modeling, a Gaussian dispersion model was used over a 32x32 km$^2$ domain (Fig. 2, right) with a spatial resolution of 200x200 m$^2$. This model is part of the Airviro system (Airviro, 2018) and incorporates a diagnostic wind model (Danard, 1977) that takes into account surface roughness and building heights, so that the model output over areas with high buildings will reflect only the conditions at roof level (i.e., the urban background). The wind model assumes that small scale

winds can be seen as a local adaptation of large scale winds (free winds) due to local fluxes of heat and momentum at the surface. The free wind is estimated from a vertical profile at the location of a meteorological station, using scaled stability variables. For this application, the input meteorological data was measured at one location (MET station in Fig. 1, left) and the building heights are illustrated in Fig. 2, right. The regional model output could be directly added to the urban model without double-counting, since the sources within the Curitiba municipality were excluded in the regional model.

The street canyon simulations at the MD monitoring site were performed with the OSPM model (Berkowicz, 2000), one of the models available in the Airviro system. This street canyon model consists of two parts, one direct plume model, following the estimated wind direction at the bottom of the street canyon. The second model part takes care of the contribution from the vortex-like re-circulation created by the surrounded buildings and is calculated by a simple box model. Neutral stability is assumed within the street canyon. The OSPM model was only used to determine the air pollution

contribution from local traffic inside the MD street canyon, to be compared to the measured increment from the roof to street level. Building heights and the dimensions of the street canyons were determined using 3D Lidar data provided by the Curitiba municipality. Private traffic volumes passing this station were estimated by municipal traffic experts as 24,075 vehicles per day, as an average over the week, composed by 93% of cars, 5% light duty diesel vehicles and 2% heavy duty diesel trucks. The public transport consisted of 469 buses per day.



## 3 Results

### 3.1 $PM_{10}$ and $NO_2$ concentrations as registered by the official monitoring network

The highest $PM_{10}$ concentrations during the three year period between 2013 and 2015 were recorded at station CIC (Table 4, note the low data capture at this station), located close to the industrial area and the ring road, with both mean and maximum

daily values within NAAQS for Brazil (50 and 150 µg m$^{-3}$, respectively). The same applies to the $NO_2$ annual mean, for which the limit value is 100 µg m$^{-3}$ and the maximum measured annual mean was 26.5 µg m$^{-3}$ at station PAR, located at a square in the city center. Averaged over the three years, all stations showed the highest values in August, which incentivized the $PM_{2.5}$ and BC monitoring campaign of this study to be performed between July and August 2016.

The ratios of the mean $PM_{10}$ and $NO_2$ levels monitored at the official network during the monitoring campaign (in August

2016) to their respective mean values in August of 2013, 2014 and 2015 were 70% for $PM_{10}$ at PAR, 50% for $PM_{10}$ at BOQ, 47% for $NO_2$ at PAR and 86% for $NO_2$ at STC, while there were no $PM_{10}$ data reported from station CIC during the campaign period. These ratios show considerably lower pollution levels during the field campaign, as compared to the average for this period during the previous three years.

While the number of public buses in operation decreased with 6% between 2013 and 2016, the official number of registered

private vehicles in Curitiba shows an increase by 4% (http://www.detran.pr.gov.br). Although the registered fleet does not necessarily equal the actual number of vehicles in circulation, it gives the best estimate of the private traffic tendency. Relating the industrial emissions, and despite the lack of data that could sustain an analysis of the trend, to the authors' knowledge no significant changes occurred over the reported 4-year period. Therefore local emissions in Curitiba are likely not to have changed significantly from the period 2013-2015 to 2016 and the identified differences in pollution levels can

most likely be attributed to variations in meteorological conditions and the long-range transported pollution arriving to Curitiba. A comparison of meteorological conditions showed considerably more precipitation during August 2016 (163 mm) compared to what is expected at this time of the year based on a 30-year climatology (73 mm), while temperature, wind direction and speed were similar to what has been registered during the last 10-20 years (INMET, 2018).

### 3.2 BC, $PM_{2.5}$ and $NO_x$ concentrations measured during the 2016 campaign

On average, BC concentrations were the highest at street level (5.5 µg m$^{-3}$), followed by rooftop level (2.3 µg m$^{-3}$, representing urban background conditions), and 2.2 µg m$^{-3}$ at SC site. The mean street canyon increment, which is defined as the difference between street and roof levels, equaled 3.2 µg m$^{-3}$ and can be attributed to local traffic along the street canyon. Fig. 3 shows the BC daily cycle in the city center with peak values occurring in the morning (ca. 10 µg m$^{-3}$) and evening (ca. 7 µg m$^{-3}$) rush hours.

Fig. 4 summarizes the descriptive statistics of BC concentrations measured in the city center (street and rooftop levels in MD) and at the SC site. The urban background concentrations in MD roof were fairly similar to those recorded at the residential site (SC), while BC levels were raised inside the street canyon (MD street). Fig. 5 shows the strong co-variation



of BC levels (r = 0.68 for hourly data) registered 13 km apart and also the peak values occurring both on weekdays and weekends. Note the high BC levels occurring during the two weekends of 6-7 and 13-14 August, not found during the previous weekend 30-31 July. After analyzing the available air pollutant data for the two high peak weekends, as well as wind direction and speed, it seems as it was the very low wind speed that made it possible for BC and PM emissions taking

place inside or just outside the city, to be advected in high concentrations over the central (MD) and southern (SC) parts of the city. Although the available information does not allow the determination of the source (or the sources), we can rule out the local traffic – which can be assumed to be a major source to BC levels during weekdays, but not during weekend nights - and long-range transport events from remote sources – which should raise the pollution levels in all stations in a more homogeneous way - as the main reason to the high peak values during the two weekends.

Fig. 6 displays boxplots of BC data gathered simultaneously in the city center with mobile monitoring and at street level at MD site. The median BC within the canyon was larger than the one obtained in the mobile samplings (5.5 and 4.0 µg m$^{-3}$, respectively). However, the mobile data showed a larger variability and extreme concentrations, with lower 5$^{th}$ percentile and larger 95$^{th}$ percentile, illustrating the heterogeneity of BC concentrations, i.e., the large variations in traffic intensity and street layout (ventilation) characterizing the biking routes across the city center.

Filter sampling of daily PM$_{2.5}$ took place both at MD rooftop and street levels. At rooftop level, the mean PM$_{2.5}$ value over the campaign period was 7.3 µg m$^{-3}$. However, the data measured at street level were affected by a technical failure and could not be used. During the mobile monitoring with the bikes, two 15-min records from weekdays' morning and afternoon rush hours were obtained during stops at the MD street level station. Those data indicated a mean PM$_{2.5}$ to BC ratio of 3. Using this scale factor on the BC data obtained at street level (mean concentration of 5.5 µg m$^{-3}$ over the period), a PM$_{2.5}$

concentration of 16.5 µg m$^{-3}$ was estimated for the street level.

At the residential site SC, the gravimetric analysis of 17 filters yielded a mean daily PM$_{2.5}$ concentrations of 36.2 µg m$^{-3}$, of which 11.2 µg m$^{-3}$ was OC and 2.6 µg m$^{-3}$ EC. This means that the total carbon (EC+OC) contributed 38% to the PM$_{2.5}$ mass, and yielded a mean OC/EC ratio of 4.4. The mean hourly PM$_{2.5}$ concentration registered by the DustTrak monitor during 25 days, after calibration against the 17 filter samples, was 25.3 µg m$^{-3}$. The longer time period for the PM$_{2.5}$ mean

value obtained for the DustTrak monitor makes it more representative than the filter mean.

The NOx concentrations at MD street site were 43 and 55 µg m-3 at roof level, and 105 and 122 µg m$^{-3}$ at street level, for the two 14-day periods of the passive samplers. From this, the street canyon increment was determined as 62 and 67 µg m$^{-3}$, respectively, for the two time periods.

### 3.3 Long-range transport as simulated by the regional model

The impact of sources outside the Curitiba municipality, including both nearby industrial sources in the Araucária municipality southwest of Curitiba, as well as remote contributions from the state of Paraná and the rest of the continent, was simulated by the regional model BRAMS 5.2 (see Section 2.5). Anthropogenic emissions, including the nearby industries, largely contributed to PM$_{2.5}$ levels, while biomass burning presented a small impact (Table 5). Since the IAP inventory did



not include BC emissions, they were taken from the coarse resolution global EDGAR-HTAP database. Consequently the BC contribution from the industrial emissions just outside Curitiba is likely underestimated by the model.

### 3.4 Simulations of local traffic impact inside the street canyon

We ran the street canyon dispersion model OSPM separately for public transport and private vehicles for the period from
July 25 to August 24, 2016, with output at the MD site. Table 6 shows the comparison of measured and simulated NOx, PM2.5 and BC impact of local traffic inside the street canyon, using emission factors from the literature (Tables 2 and 3). For NOx, there was a relatively good agreement, but for both $PM_{2.5}$ and BC the simulated impact was much lower than what the measurements indicate. Since uncertainties related to the meteorological data and model assumptions are equal for all three compounds, it is reasonable to assume that the larger uncertainties found for BC and $PM_{2.5}$ are caused by uncertainties
in the effective emission factors. In addition, for $PM_{2.5}$, there is a fraction related to road, tire and break wear, which should be added to the exhaust emission factors used in the model simulation. To some degree the underestimated effective emission factors for the private vehicles may be related to fleet composition (for the public transport the fleet composition was known in detail). However, heavy duty vehicles that do not form part of the public transport are restricted in the studied area, thus reducing this source of uncertainty.
Since BC measurements were available as hourly data, it was possible to perform a multiple regression analysis with measured BC increment as dependent variable and the two simulated hourly time series of the contributions from buses and private traffic as independent variables. The regression analysis suggests that the public transport signal (bus impact) should be multiplied by a factor of 1.2 and the private traffic by a factor of 5. Fig. 7 and the mean values of Table 7 show that with these corrected emission factors, the model output matches the measured street increment in the street canyon. With the
assumed fleet composition, the effective BC emission factor for the mixed private vehicle fleet was corrected from 4 to 19 mg veh$^{-1}$ km$^{-1}$.

There were no hourly $PM_{2.5}$ data to perform a similar regression analysis; however, scaling up the summed model output of the bus and private vehicle contributions to $PM_{2.5}$ by a factor of 5 yielded a local contribution similar to the monitored street canyon increment.

### 3.5 Urban and regional simulations of $PM_{2.5}$ and BC

The Gaussian urban model was applied with the corrected emission factors for the on-road vehicles, and the results were subsequently added to the regional LRT contributions. Table 8 shows the contributions at the two sites and Fig. 9 displays the mean spatial distribution of the simulated $PM_{2.5}$ and BC concentrations. The emission inventory show local $PM_{2.5}$ emissions from the transport sector (public and private together) within the urban model domain over Curitiba of 643 tons
year$^{-1}$, which can be compared to 329 tons year$^{-1}$ from the smaller industries located inside the city and 1,600 tons year$^{-1}$ (calculated as 70% of officially reported $PM_{10}$ emissions) from the industrial Araucária area just southwest of the city. For




BC, we estimated 18 tons year$^{-1}$ from public transport and 375 tons year$^{-1}$ from private traffic (no information was available on the industrial contribution).

## 4 Discussion and conclusions

The information on $PM_{10}$ and $NO_2$ levels from the official monitoring network in Curitiba revealed mean annual $PM_{10}$

concentrations between 15 and 30 µg m$^{-3}$, which are between the values presented in the WHO global assessment for high-income countries and for low- to middle-income countries in the Americas (see Fig. 3 in WHO, 2016). Comparatively, cities of the size of Curitiba in the Eastern Mediterranean and South-East Asia have much higher annual mean concentrations, spanning from 100 to 200 µg m$^{-3}$.

The monitoring campaign was only a month long and planned to be representative of wintertime conditions in the southern

hemisphere, when the highest pollution levels are usually observed. However, the meteorological conditions, and likely also a smaller contribution from long-range transport than usual, caused the $PM_{10}$ and $NO_2$ levels to be considerably lower during the 2016 field campaign, as compared to average concentrations in previous years. This indicates that monitored $PM_{2.5}$ and BC concentrations reported for the 2016 winter month campaign could also have been lower than what is normally the case for the months of July and August, likely more comparable to annual mean values of full years as given by earlier studies.

The urban background $PM_{2.5}$ concentrations registered at the roof top in the city center (7.3 µg m$^{-3}$) agreed reasonably well with the 10.3 µg m$^{-3}$ reported from a year-long measurement in 2014-2015 at the Federal University campus (Polezer et al., 2018), considering the fact that the station was located approximately 100 m from an interstate highway. Measurements conducted in 2007 and 2008 at the same location in the campus showed a slightly higher mean $PM_{2.5}$ concentration of 14.4 µg m$^{-3}$ (De Miranda et al., 2012; Andrade et al., 2012) possibly reflecting higher vehicular emissions at that time.

At the residential station SC, the mean $PM_{2.5}$ concentration (25.3 µg m$^{-3}$) was significantly higher than in the urban background of the city center (7.3 µg m$^{-3}$). The existence of an additional local source of $PM_{2.5}$, not included in the emission inventory, can be deduced from the mapped $PM_{2.5}$ concentrations of Fig. 8 (left), which does not indicate any levels in the range of 20-30 µg m$^{-3}$ around the SC station. The mean OC/EC ratio of 4.4 (range 3.3-9.0) found in $PM_{2.5}$ sampled at SC indicates that the dominant local emissions in this area were unlikely to originate from diesel traffic emissions along the

nearby ring road, since the EC contribution for diesel vehicles can be expected to be larger than the OC part (Harrison and Yin, 2008). Comparatively, a monitoring campaign performed at four urban sites in São Paulo (Monteiro dos Santos et al., 2016) reported OC levels between 2.65 and 3.37 µg m$^{-3}$ and EC levels between 6.11 and 1.50 µg m$^{-3}$, yielding OC/EC ratios of 1.57, 1.88, 1.89 and 0.56. The lowest OC/EC was observed within a street canyon, suggesting a large contribution from diesel-fueled vehicles. Thus, the higher OC/EC ratio found at the Curitiba residential station, makes it unlikely that traffic

emissions and the closeness to the ring road were the cause to the higher $PM_{2.5}$ levels. Instead, the high OC levels may indicate the presence of wood burning around the SC station. Curitiba municipality lacks information on such residential




biofuel devices, but the staff operating the monitoring station occasionally reported the smell of wood smoke and noted the presence of several houses with small chimneys in the neighborhood.

The simulated BC levels in the urban background, i.e., excluding street canyons and the immediate vicinity of the major roads, were fairly homogenous over the city (Fig. 8, right). Also the few existing measurement data points indicated a spatial

homogeneity, with mean values just above 2 µg m$^{-3}$ at the sites MD roof and SC reported in this paper and also at the university campus, considering the year-long dataset from 2014-2015 (Polezer et al., 2018). Within street canyons, the BC levels were locally raised (Johansson et al., 2017; Krecl et al., 2016) and also close to highways with diesel traffic (Andrade et al., 2012). Even if BC levels show to be spatially homogenous in the urban background, there were large temporal variations due to meteorological conditions, changes in the long-range contribution and likely due to some unidentified BC

sources inside or just outside the city. The temporal evolution of BC levels at station MD and SC in Fig. 5 shows high BC levels at both stations during the second and third weekends of the monitoring campaign, however, considerably higher BC peaks were observed at the residential site SC. These enhanced BC peaks at the SC station may have resulted from wood combustion in the vicinity of the SC station, even if the BC levels were not increased as much as the OC and PM$_{2.5}$ levels.

The mean BC concentration of 5.5 µg m$^{-3}$ observed at MD street level was similar to what was registered during weekdays in

a Stockholm street canyon in 2006 (5.1 µg m$^{-3}$). A later comparison in 2013 for the same site in Stockholm showed reduced levels (2.2 µg m$^{-3}$) explained by technologically improved vehicles trafficking this street canyon (Krecl et al., 2017). This gives an indication of what is possible to achieve with regulations on vehicle technology.

The regional model was used to estimate the long-range contributions of PM$_{2.5}$ and BC to the city. The emission inventory of industrial sources only included PM$_{10}$ emissions, which required an assumption that 70% of those were in the fine particle

size range. Table 5 thus gives the magnitude of the PM$_{2.5}$ impact from industrial sources and also the PM$_{10}$ impact, which can be seen as an upper limit if all industrial PM emissions are found in the PM$_{2.5}$ fraction. The regional model also indicated a rather small (0.5 µgm$^{-3}$) contribution of biomass burning to PM$_{2.5}$ in Curitiba for the monitoring campaign period. Unfortunately, the regional model did not have access to complete BC emission data, since the Paraná inventory of industrial emissions only covered PM$_{10}$. With only the large-scale global EDGAR-HTAP emissions, the BC and OC content in the

long-range transported air arriving to Curitiba was clearly underestimated by the model.

An important aim with the ParCur assessment in Curitiba was to determine the emission of PM$_{2.5}$ and BC from sources inside the city. For that purpose, the street canyon measurements were used to evaluate in situ the emission characteristics of the vehicle fleet in Curitiba. With emission factors taken from the literature as input to the dispersion model, there was good agreement between the model outputs and the measured NOx concentrations, but the BC increment within the street canyon

was underestimated by a factor of 3 (Table 6). The OSPM model has been used and evaluated in many urban environments and an extensive comparison showed good results for NO$_x$ (Ketzel et al., 2012). If assumed that the model gives reasonable results, then the good resemblance between simulated and monitored NO$_x$ values indicates that the NO$_x$ emission factors should be fairly accurate, but also that the BC emission factors are erroneous. With this as motivation, new emission factors for BC were determined through regression analysis. An interesting fact is that the regression analysis confirmed the




magnitude of the emission factors of the buses, whose fleet composition, age and equivalent Euro class we know in detail. However, the regression pointed out a large underestimation (factor of 5) of the mixed private vehicle fleet emissions, for which the composition and technical status was much less known. A factor of 5 of difference in an emission factor may be seen as a too large discrepancy. However, BC emission factors show a large scatter and both the original 4 mg veh$^{-1}$ km$^{-1}$ and

the corrected 19 mg veh$^{-1}$ km$^{-1}$ for the mixed private vehicle fleet circulating through the MD street canyon fit into "real-world" emission factors reported in the literature. Through chasing individual vehicles, BC emission factors spanned from 10 to 32 mg BC km$^{-1}$ for gasoline cars (Ježek et al., 2015), while for heavy-duty diesel vehicles they went from 53 mg BC km$^{-1}$ (Park et al., 2011) to values 10-fold higher (Wang et al., 2012). An assessment of real-world BC emission factors performed in a street canyon in Stockholm (Krecl et al., 2017) for the year 2006 (when BC levels within the street canyon were similar

to those reported for M. Deodoro in Curitiba) found a BC emission factor for gasoline cars of 11 mg veh$^{-1}$ km$^{-1}$, which is significantly higher than the 0.15 mg veh$^{-1}$ km$^{-1}$ used originally in the present study (Table 3) and can therefore justify the higher emission factor coming out of the regression.

For PM$_{2.5}$, which at the MD station was only measured as daily averages, it was only possible to compare the simulated total impact of all vehicles in the street canyon with the measured increment (i.e., the difference between the more polluted air in

the bottom of the street canyon and the cleaner atmosphere at the rooftop). Also here the simulated PM$_{2.5}$ contribution of the local traffic was largely underestimated when compared to the measured increment. For PM$_{2.5}$, contributions from non-exhaust particles generated by the vehicles are also expected, namely those consisting of wear particles from brakes and tires, together with wear of the street pavement itself, which were not included in the emission factors of Tables 2 and 3. However, since the aim was to estimate the total PM$_{2.5}$ emissions, no separation was made between exhaust and non-exhaust

PM$_{2.5}$ originating from the traffic in Curitiba. Since it is not possible to perform a regression that can apportion the impact on PM$_{2.5}$ of public transport and private vehicles (lack of hourly data), a general correction with a factor of 5 was applied to both buses and the mixed fleet of private vehicles.

The boxplot comparison between mobile measurements of BC along the four biking trajectories in the city center and the measured BC levels at the fixed MD street station revealed the large heterogeneity in traffic rates and street layouts in the

city center when compared to a fixed site in a busy canyon street. The variability in traffic rates create differences in emission rates at microscale level (individual street canyons), and varying building heights will also influence the immediate dilution of air pollutants emitted by local traffic.

A goal of the ParCur project was to map the annual concentrations of PM$_{2.5}$ and BC over the city, identifying the contribution of local sources and the magnitude of the long-range transport. As indicated by Table 8, a complete picture has

not been possible to achieve. For PM$_{2.5}$, the summed model output attained 5.1 µg m$^{-3}$ compared to 7.3 µg m$^{-3}$ measured in the city center. The missing 30% contribution could possibly be attributed to sources not included in the emission inventory (e.g. restaurants and residential sources), and also to underestimated particle mass in the regional model output (which did not include secondary aerosol formation). At the residential site, the simulated PM$_{2.5}$ levels only covered 20-25% of the





measured concentrations, indicating a strong local source not being included. As discussed earlier, there are indications of wood combustion taking place in this residential area on the city's outskirts, but not in its center.

For BC, the simulated concentrations constituted 56% of the measured concentrations in the city center and exactly half at the residential location. Most of this difference is likely due to a too low BC concentration in the simulated long-range

contribution, since the emission inventory lacked industrial emissions of BC (e.g. those in Araucária just southwest of Curitiba). For the residential site, the existence of residential wood combustion would also contribute to increased BC levels. However, not as large as for $PM_{2.5}$, this since high OC/EC ratios are expected in fumes coming from wood combustion.

The few monitoring points and the short sampling period contributed to certain uncertainties in our conclusions concerning the $PM_{2.5}$ and BC emissions and concentrations in Curitiba. For the city's inner core, the information from the fixed MD

station and the mobile monitoring, supported by the model results, should together allow a good description of emissions and concentrations. However, the higher air $PM_{2.5}$ and BC concentration indicated in the model output (Fig. 8) along the ring road "Contorno" and around the industrial area southwest of Curitiba, could not be verified by local monitoring and also included various methodological simplifications, such as the vehicle fleet composition, emissions from the industries, that makes the concentrations indicated for this area more uncertain. In connection to this problem, great uncertainty subsists in

knowing the actual characteristics of the gross polluters, notably old trucks (Fig. 9) in transit through Curitiba's ring road. Many of them are not registered in the state of Paraná and, consequently, not taken into account in our inventory.

The outcome of the monitoring campaign was positive for the BC part, whereas the $PM_{2.5}$ sampling at the central street canyon station implied two difficulties. The major one was the technical problems that impeded the calculation of the increment from roof to street and the necessity of using BC data as a proxy for $PM_{2.5}$. Also, the fact that $PM_{2.5}$ data had daily

resolution made it difficult to separate public transport and private vehicles, which had different temporal variation during the day. Since the determination of local emissions and their impact rely heavily on monitored data, it is fair to conclude that the BC results, both monitored and simulated, should be seen as more robust than the $PM_{2.5}$ results.

This study highlights the need to develop emission inventories for $PM_{2.5}$ and BC at local, regional and national levels that are currently missing in Brazil. Other local activities such as small combustion from backyard burning and wood burning

appliances, and emissions from pizzerias, bakeries and steakhouses should be also considered in the inventories.

Data on airborne fine particle emissions and concentrations in cities are valuable to traffic and air quality managers, urban planners and landscape architects, health practitioners, researchers, and ultimately to legislators and decision makers. Most Brazilian cities lack this kind of information which hinders an accurate assessment of their ambient concentrations, the source apportionment and the potential health outcomes. Air quality monitoring networks with sufficient spatial coverage are

unlikely to be available in Brazil in the near future. Hence, methods, instruments and models to identify air pollution hotspots and their sources are an urgent need to safeguard the population exposure to toxic airborne species. This paper, as a result of the Brazil/Sweden cooperation materialized in the ParCur project, addressed this gap taking Curitiba as case study. The integrated approach used in the study can be implemented in other Brazilian cities as long as an open data policy and a close cooperation between municipal authorities and academia can be achieved.





**Author contributions**

L Gidhagen, P Krecl, A Créso Targino, R Godoi and J H Amorim have outlined the structure and draft texts of this manuscript. All co-authors have contributed with text revisions, technical details within their specific area of responsibility

and the discussion of the results. The specific areas were: L Gidhagen (local dispersion models), P Krecl (monitoring fixed station Sítio Cercado), A Créso Targino (mobile monitoring), G Polezer and R Godoi (monitoring fixed station M. Deodoro), F Castelhano and F Mendonça (meteorological data and analysis), E Felix (monitoring passive samplers, filter weighting), Y Cipoli (mobile data collection and data analysis), F Malucelli (private traffic data as model input), A Wolf (public transport data as model input), M Alonso (regional dispersion model), D Segersson (emission inventory as model input), J H Amorim

(design and execution monitoring campaign).

**Funding sources**

This study has been supported on the Swedish side by the Swedish Ministry of Environment and Energy, and from the Brazilian counterpart by the Federal University of Paraná, the Federal University of Technology of Paraná, the Federal University of Pelotas and the Municipality of Curitiba. The instrumentation for the mobile sampling was acquired with funds

of grant 404146/2013-9 from the National Council for Scientific and Technological Development of Brazil (CNPq).

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





Fig. captions

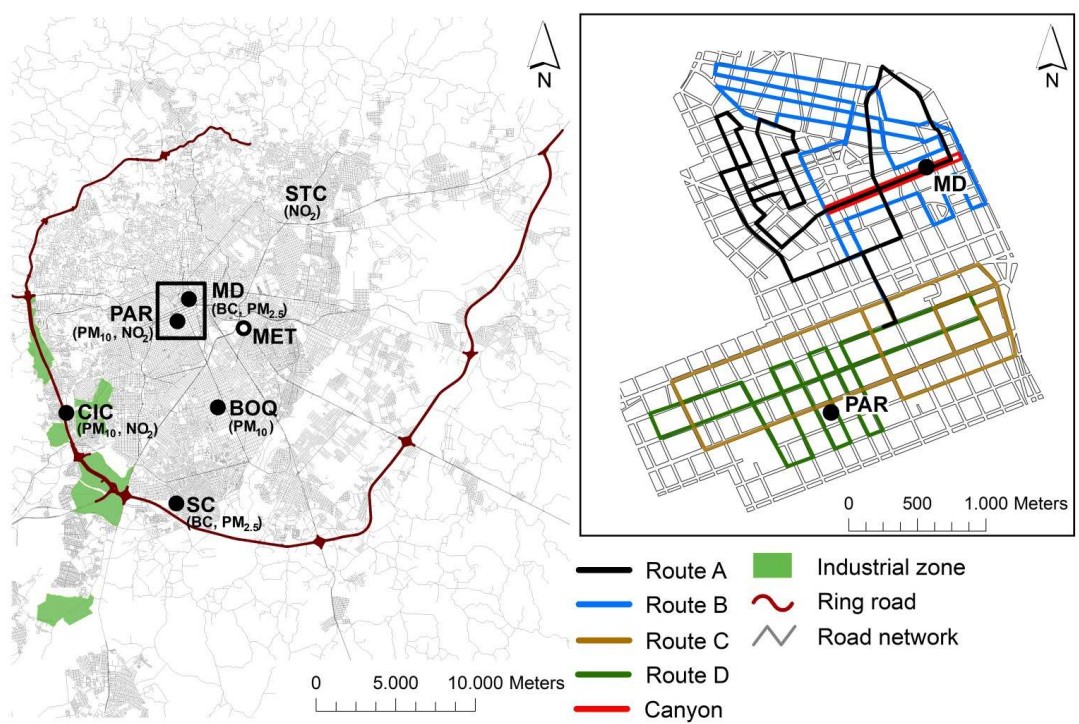

**Fig. 1:** Maps showing the location of: i) Left: the four official monitoring stations (CIC, PAR, BOQ, STC), the site in the residential area (SC), and the meteorological station (MET), ii) Right: the street canyon site (MD), and the four biking trajectories used for the mobile measurements.



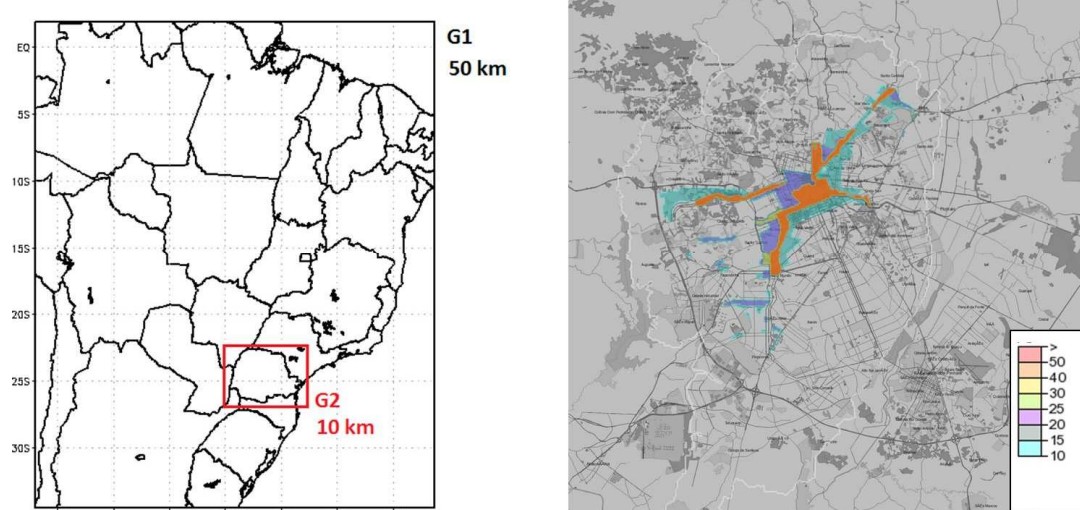

**Fig. 2:** Domains for the regional model (left) and the urban dispersion model (right). For the latter, building heights above 10 m are marked (right). Over the remaining built-up city areas the building height was set to 6 m.





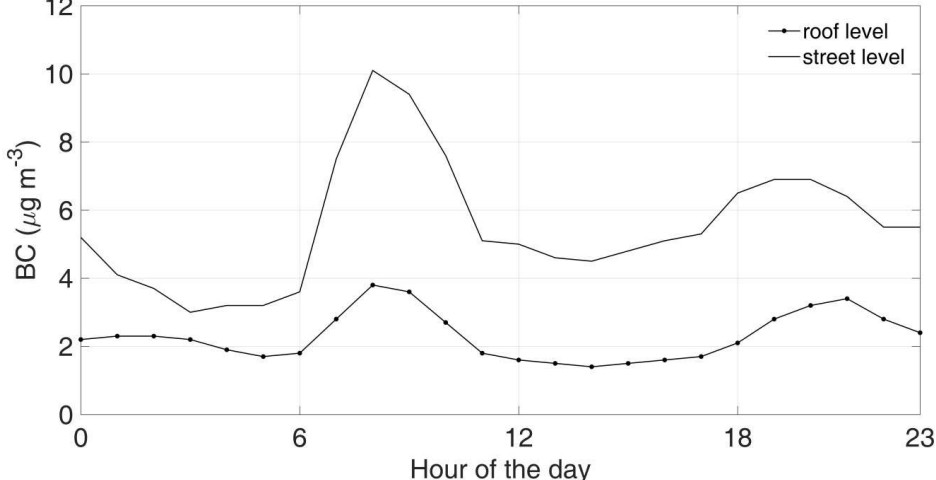

**Fig. 3:** Mean daily cycle of BC concentrations (at 880 nm wavelength) measured at roof and street levels at MD site. Period: July 25 – August 24, 2016 (all days of the week).





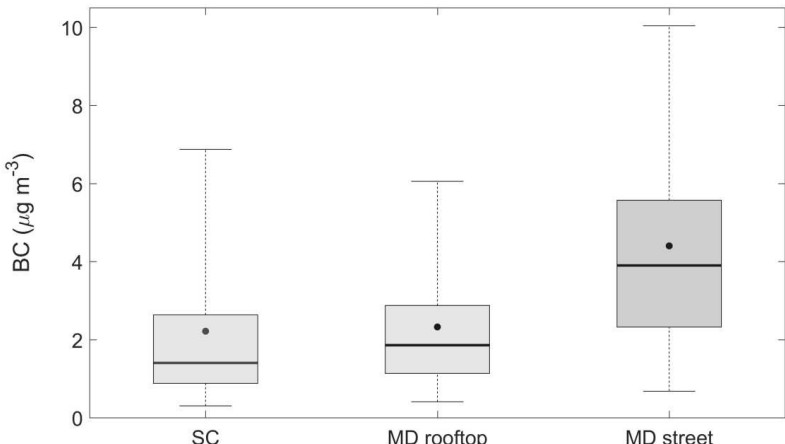

**Fig. 4:** Boxplots of hourly BC concentrations measured at MD and SC sites in the period July 25 – August 24, 2016. The midline is the median, the upper and lower limits of the box are the 75th and 25th percentiles, the whiskers are the 5th and 9th percentiles, and the black dot is the mean.





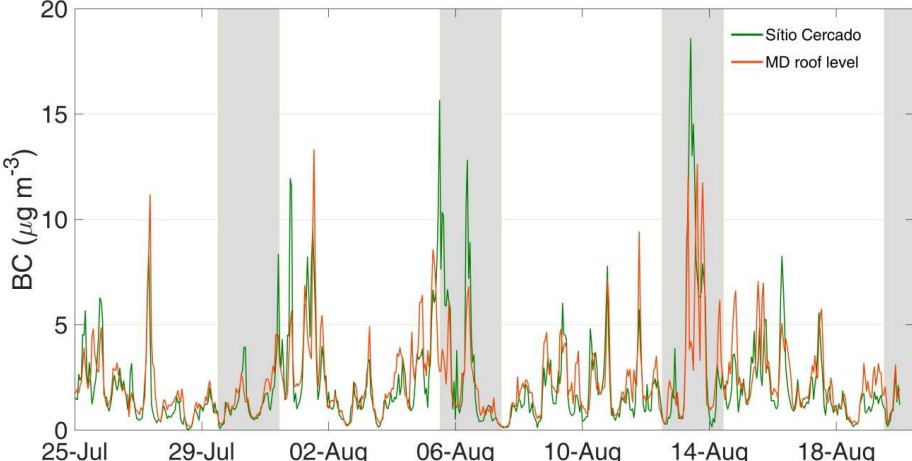

**Fig. 5:** Time series of hourly BC concentrations registered at MD rooftop and at SC from July 25 to August 20, 2016. The shaded areas represent the weekends.





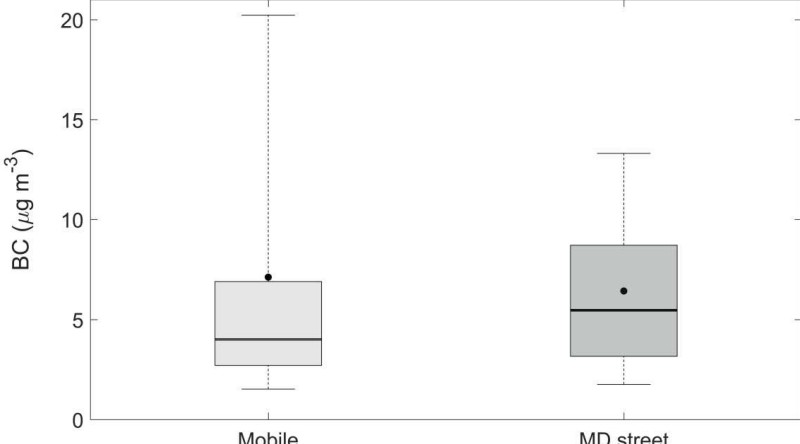

**Fig. 6:** Boxplot of BC data collected with mobile monitoring (10-s resolution) along the four biking routes and from 5-min concentrations simultaneously measured at MD street site. The midline is the median, the upper and lower limits of the box are the 75$^{th}$ and 25$^{th}$ percentiles, the whiskers are the 5$^{th}$ and 95$^{th}$ percentiles, and the black dot is the mean.



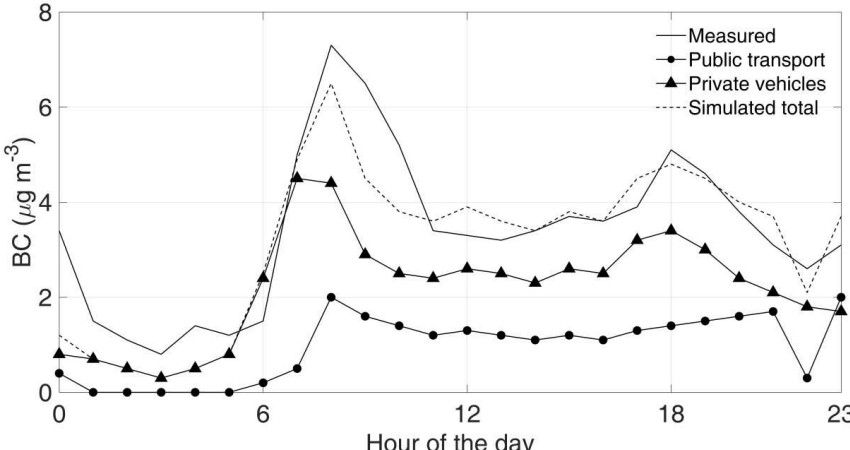

**Fig. 7:** Daily mean variation of measured and simulated BC contributions from local traffic at MD street site. The simulated impact was corrected with a factor of 1.2 for public transport and a factor of 5.0 for private vehicles.




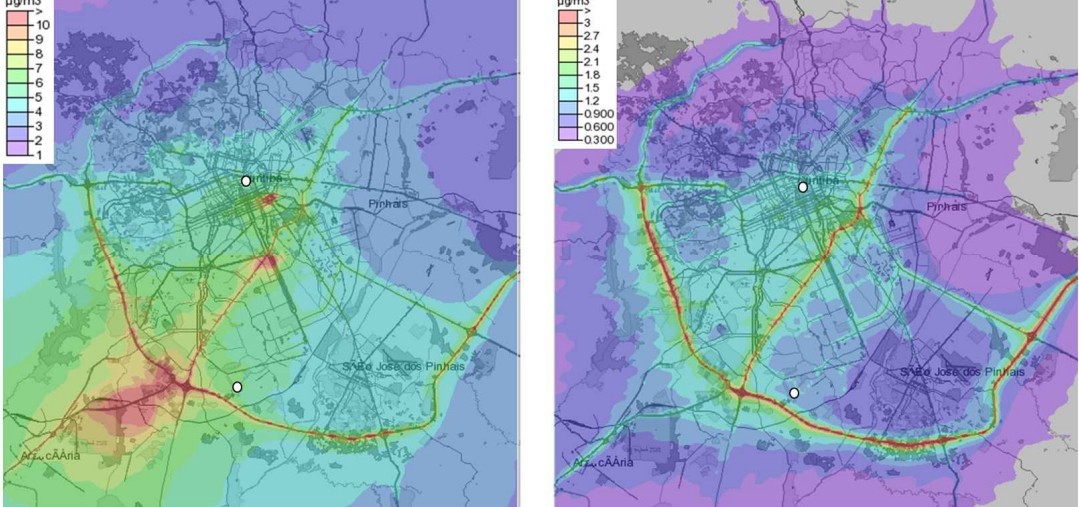

**Fig. 8:** Modelled mean concentrations of PM$_{2.5}$ (left) and BC (right) in the period from the 25$^{th}$ of July to the 24$^{th}$ of August, 2016. Regional (anthropogenic and biomass) and local (industry inside Curitiba, public transport and private vehicle impact) contributions were included. The city center (MD) and the residential (SC) stations marked as white circles.





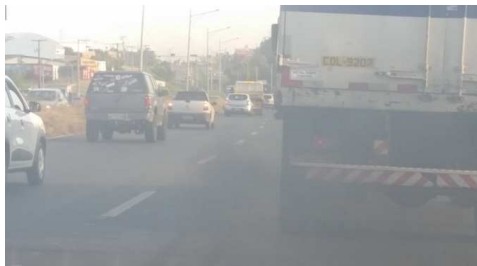

**Fig. 9:** Photo of a truck circulating in Curitiba's ring road, illustrating the intense exhaust emissions.



**Tables**

**Table 1.** Number of buses in the public transportation system of Curitiba classified by type, technology and fuel type for the year 2016 (source: Municipality).

| Bus type and length | Euro II diesel | Euro III diesel | Euro III biodiesel | Euro V Diesel | Euro V biodiesel | Total |
|---|---|---|---|---|---|---|
| Micro (8 m) | 18 | 2 | 0 | 0 | 0 | 20 |
| Special micro (10 m) | 0 | 172 | 0 | 0 | 0 | 172 |
| Common (12 m) | 52 | 555 | 0 | 3 | 0 | 610 |
| Semi-standard (13 m) | 0 | 30 | 0 | 0 | 0 | 30 |
| Standard (13 m) | 9 | 328 | 0 | 0 | 0 | 337 |
| Standard hybrid (13 m) | 0 | 0 | 0 | 28 | 2 | 30 |
| Articulated (18.6 m) | 38 | 198 | 0 | 0 | 0 | 236 |
| Articulated (20 m) | 0 | 35 | 6 | 0 | 0 | 41 |
| Bi-articulated (25/28 m) | 54 | 92 | 26 | 0 | 0 | 172 |
| **Total** | **171** | **1424** | **32** | **31** | **2** | **1660** |



**Table 2.** Emission factors for public transport.

| Bus classes | Technology | NO$_x$ (mg veh$^{-1}$ km$^{-1}$) | PM exhaust diesel (mg veh$^{-1}$ km$^{-1}$) | PM exhaust biodiesel (mg veh$^{-1}$ km$^{-1}$) | BC diesel (mg veh$^{-1}$ km$^{-1}$) | BC biodiesel (mg veh$^{-1}$ km$^{-1}$) | Fuel cons. (ml veh$^{-1}$ km$^{-1}$) |
|---|---|---|---|---|---|---|---|
| Micro | Euro II | 9840 | 184 | 92 | 120 | 60 | 326 |
| Standard | Euro II | 13080 | 264 | 132 | 172 | 86 | 444 |
| Articulated | Euro II | 16380 | 373 | 187 | 242 | 121 | 568 |
| Bi-articulated | Euro II | 19438 | 443 | 221 | 288 | 144 | 674 |
| Micro | Euro III | 9020 | 171 | 86 | 120 | 60 | 344 |
| Standard | Euro III | 11740 | 237 | 119 | 166 | 83 | 463 |
| Articulated | Euro III | 14770 | 285 | 143 | 200 | 100 | 588 |
| Bi-articulated | Euro III | 17527 | 338 | 169 | 237 | 118 | 698 |
| Micro | Euro V | 6690 | 52 | 26 | 39 | 20 | 298 |
| Standard | Euro V | 8370 | 68 | 34 | 51 | 26 | 410 |
| Articulated | Euro V | 7750 | 81 | 41 | 61 | 30 | 535 |
| Bi-articulated | Euro V | 9197 | 96 | 48 | 72 | 36 | 634 |



**Table 3.** Emission factors for private vehicles were taken from the report by the European Environment Agency (EEA, 2016), where BC emission factors are expressed as percentages of PM$_{2.5}$ emissions (see Table 3-91 in EEA, 2016). Cars had flex-fuel engines running predominantly on gasoline. Heavy duty vehicles (HDV) constitute a mix of emissions standards, 5    here simplified to an intermediate technology of Euro III.

| | | NO$_x$ (mg veh$^{-1}$ km$^{-1}$) | PM exhaust (mg veh$^{-1}$ km$^{-1}$) | BC (mg veh$^{-1}$ km$^{-1}$) | Fuel cons. (ml veh$^{-1}$ km$^{-1}$) |
|---|---|---|---|---|---|
| Gasoline cars | Euro 4 | 61 | 1 | 0.15 | 115 |
| Diesel LDV, < 3 tons | Euro 4 | 831 | 41 | 36 | 95 |
| Diesel HDV, 16-32 tons | Euro III | 6270 | 130 | 91 | 250 |



**Table 4.** $PM_{10}$ and $NO_2$ concentrations from Curitiba's official monitoring network for the period 2013-2015. Unit: µg m$^{-3}$.

| Pollutant | Station | Mean | Hourly max | Daily max | Month with highest concentration | Data capture |
|---|---|---|---|---|---|---|
| $PM_{10}$ | PAR | 15.1 | 180 | 86 | August | 91% |
| | CIC | 30.3 | 326 | 120 | August | 43% |
| | BOQ$^*$ | 14.5 | 197 | 90 | August | 88% |
| $NO_2$ | PAR | 26.5 | 201 | 89 | August | 88% |
| | CIC | 22.5 | 148 | 57 | August | 59% |
| | STC$^*$ | 13.2 | 90 | 43 | August | 76% |

$^*$ At BOQ only $PM_{10}$ was measured and at STC only $NO_2$.



**Table 5.** Contribution of long-range transported pollution (LRT) to MD in the city center and at the residential site SC. Period: July 25th to August 24th, 2016.

| Emission sources | | MD rooftop ($\mu g\ m^{-3}$) | SC ($\mu g\ m^{-3}$) |
|---|---|---|---|
| Anthropogenic including IAP inventory | $PM_{10}$ | 2.4 | 4.3 |
| Anthropogenic including IAP inventory | $PM_{2.5}$ | 1.7* | 3.0* |
| 3BEM biomass burning | $PM_{2.5}$ | 0.5 | 0.5 |
| EDGAR-HTAP | BC | 0.06 | 0.06 |
| EDGAR-HTAP | OC | 0.46 | 0.46 |

5   * The IAP inventory provided the $PM_{10}$ emissions, anthropogenic $PM_{2.5}$ emissions were estimated as 70% of $PM_{10}$ (see Section 2.3)).



**Table 6.** Comparisons between simulated and measured air pollutants from local traffic inside MD street canyon. Note that the measurements are reported here as street increments (given as the difference between street and roof level concentrations) to be comparable with the model output, which only gives the impact of the road traffic inside the street canyon.

| Period | | Measurements ($\mu g\ m^{-3}$) | Model simulation ($\mu g\ m^{-3}$) | | |
|---|---|---|---|---|---|
| | | | Buses | Private vehicles | Total |
| $29^{th}$ July – $12^{th}$ August, 2016[*] | $NO_x$ | 62 | 43 | 32 | 75 |
| $15^{th}$ July – $29^{th}$ August, 2016[*] | $NO_x$ | 66 | 40 | 29 | 69 |
| $25^{th}$ July – $24^{th}$ August, 2016 | $PM_{2.5}$ | 9.3[*] | 0.9 | 0.7 | 1.6 |
| $25^{th}$ July – $24^{th}$ August, 2016 | BC | 3.2 | 0.6 | 0.5 | 1.1 |

[*] $PM_{2.5}$ increment estimated by using a $PM_{2.5}$ to BC ratio of 3 at street level, based on mobile measurements (see previous comment of $PM_{2.5}$ measurements at MD street site).



**Table 7.** Measured street increments and simulations of local traffic impact inside the canyon where the monitor station MD (street level) was located, after correcting the emissions factors.

| Period | | Measurements ($\mu g\ m^{-3}$) | Model simulation ($\mu g\ m^{-3}$) | | |
|---|---|---|---|---|---|
| | | | Buses | private | Total |
| 25th July – 24th August, 2016 | PM$_{2.5}$ | 9.3 | | | 8.0[*] |
| 25th July – 24th August, 2016 | BC | 3.2 | 0.8[**] | 2.2[**] | 3.0[**] |

5   [*] Total simulated PM2.5 output of Table 6 corrected with a factor of 5.
[**]Bus impact of BC corrected with a factor of 1.2 and private traffic impact of BC corrected by a factor of 5.0, with values given by a regression analysis of hourly data.



**Table 8.** Simulated contributions of regional and local sources inside Curitiba (urban model output) to ambient concentrations at the MD rooftop and SC sites. Period: 25[th] July to 24[th] August, 2016. Unit: $\mu g \ m^{-3}$.

| Contribution | model | MD roof ($PM_{2.5}$) | SC ($PM_{2.5}$) | MD roof (BC) | SC (BC) |
|---|---|---|---|---|---|
| Anthropogenic with IAP | regional | 1.7 | 3.0 | - | - |
| 3BEM biomass burning | regional | 0.5 | 0.5 | - | - |
| Anthropogenic + biomass | regional | - | - | 0.06 | 0.07 |
| Industry inside Curitiba | urban | 0.6 | 0.6 | - | - |
| Public transport | urban | 2.3[*] | 1.8[*] | 0.13 | 0.07 |
| Private vehicles | urban | | | 1.07 | 0.98 |
| **Summed model** | | **5.1** | **5.9** | **1.27** | **1.11** |
| Monitored | | 7.3 | 25.3 | 2.29 | 2.22 |

5   [*] The contribution could only be simulated as the sum of public and private traffic since the same correction factor was applied.