# Peer review of "Experimental and model assessment of PM2.5 and BC emissions and concentrations in a Brazilian city – the Curitiba case study"

_Atmospheric Chemistry and Physics, 2018_

## Referee Comment (RC1) · Anonymous Referee #1 · 6 Jan 2019

The manuscript carried out monitoring and model assessments in a Brazilian city Curitiba focusing on PM 2.5 and Black Carbon emissions. It is an interesting and important research topic, however, there are information details lacking in the current manuscript and it is like a report instead of a research paper. There is a large block of text (for example in the discussion and conclusion section) which is difficult for the readers to follow.   I would suggest a major revision before we can consider for publication.

It is suggested that the authors would improve the manuscript by adding sub-headers at appropriate locations and also organize the tabulated data in form of bar/pie charts. Below are some specific comments:

**Introduction**:

- Page 2 line 6: "One Specific SDG indicator, 11.6.2, for meeting this goal is the annual mean level of PM2.5 weighted by population…" This sentence seems incomplete. Are the authors trying to bring out that the annual mean level of population weighted PM2.5 reaches a certain value would meet the goal? Please consider re-organize the sentence structure.

**Section 2.1 Study Area:**

- Page 4 line 1: The authors mentioned there are four automatic stations within the municipality of Curitiba and four in the industrial area of the nearby city of Araucaria and an analysis of PM10 and NO2 from the official monitoring network was performed for three years from 2013 to 2015. The analysis include the data from the four automatic stations within the municipality of Curitiba but does it consider the data from the four stations in the industrial area of the nearby city of Araucaria? If not, what is the reason for excluding them in the analysis? Also please clarify whether there are missing data?

**Section 2.2 Study Design:**

- Page 4 line 7: Please be specific measured data from which stations are validating the emission inventory?

**Section 2.3 Emission Inventory:**

- The emission inventory are developed mainly based on two economic sectors: industries and on-road transport.

  - Page 5 lines 5-8: Detailed data for industrial sources are lacking. Please summarize and supplement.
  - It is observed that data has been provided in Tables 1-3 for both public and private vehicles. Please supplement bar charts to these tables for better illustration.  The data on public transportation are quite detailed but for private vehicles, it is recommended to report the total number/types of vehicles and average speed as now seem there is only emission factors for private vehicles as in Table 3.  What about uncertainty analysis?
  - Page 5 line 23: Please describe more details about the VISSIM model and the corresponding input settings and data required for this model.
  - Page 5 line 25: Please provide information on the daily profiles adopted.

**Section 2.5 Dispersion Modelling:**

- The EDGAR-HTAP information include BC and OC emissions from what kind of sources, does the information include those BC release from vehicles and industries or wood burning within state of Parana?

- It is mentioned that Gaussian Dispersion model incorporates a diagnostic wind model that takes into account surface roughness and building heights. As the building morphologies of the investigated city, are quite complex revealed in Fig 2 (right). Do the authors require inputting surface roughness and building height information or construct the CAD city model for OSPM model? If so, what surface roughness profiles/values and assumptions do the authors made to input in the OSPM model? Please include in the manuscript.

- Would the justification of choice of BRAMS for regional scale modelling (P.6 line 27-29) be provided? How would be its accuracy compare with other major alternatives e.g. WRF?

**Section 3 Results:**

- Page 7 Line 28: There were no PM10 data reported from station CIC during the campaign period → Does it mean there is no PM10 release or there are other reasons behind? How about NO2 data at this station?

- Page 7 Line 32: Would the authors please provide more information on the variation in long range transported pollution arriving to Curitiba.

- Page 8 Line 21: Please describe what kind of technical failure caused the acquired data during fixed monitoring cannot be used.

- Section 3.4: It is mentioned that the street canyon dispersion model OSPM produced much smaller magnitude of PM 2.5 and BC compared to measurement and regression analysis has been carried out to obtain correction factors. Please provide some brief information on the way that regression analysis were carried out. Also are the correction factors applicable to other places in Brazil? Have the authors tried other dispersion models such as AERMOD, ADMS or CFD approach, which might improve the magnitude prediction?

- Section 3.4: Can the change of correction factor for BC for private vehicle emission by a factor of 5 be justified and conclude that it is not contributing by other factors like wrong fleet composition and modeling error? Is the new emission factor checked using an independent dataset of concentration measurement?

**Section 4 Discussions and conclusions:**

- This section needs to be better organized around the main conclusions and highlights. At the moment, there is a large block of text with many information which is hard for the readers to follow. Please add sub-headers at appropriate locations to break down the text and also present the important data in form of bar/pie charts instead of tables only. Please also list out important conclusions point by point and include a section on limitations of the study.

- Please add a sub-section that focus on discussion of comparison between model simulation results and measurement/monitoring data.

- EDGAR-HTAP gives only 0.1deg x 0.1deg resolution. Do you mean local industrial BC emission is low when compared to other BC sources in Curitiba by providing a simulated BC urban background (P.10 line 32-33)? Please justify?

---

## Referee Comment (RC2) · Anonymous Referee #2 · 11 Jan 2019

General comments The article presents important results regarding campaigns of measures in the scope of the ParCur project. The observational part was made based on measures close to the surface, in fixed and "mobile" stations (bicycles), and at the level of 70 m above the ground. This is the richest and most important part of the article. Also, three different models were applied to try to represent the concentrations of Black Carbon, Particulate Material 2.5 and Oxides of Nitrogen. Despite three different approaches to modeling, little useful information has been obtained, making several approaches, both in terms of emissions and in terms of the results obtained, and is therefore the weak part of the article. The general recommendation is to keep only the observational part of the work, with the modeling part being excluded. Here are some

specific points that led to this conclusion.

Specific comments

Page 2, line 19: correct the units "$\mu$g m$-3$"

Page 2, line 29: Is the year of the reference correct? In the reference list it appears as 2014

Page 3, lines 17 – 20. The authors said that "As far as we know, this is a pioneering study in South America with the integration of fixed and mobile high spatio-temporal resolution PM2.5 and BC measurements, the development of emission inventories and the implementation of modeling tools at different spatial scales for validating the emissions and to determine the spatial distribution of pollutant concentrations across the city." Maybe you are referring to studies about Black Carbon over Curitiba, but, in fact, there are at least a dozen of studies combining high resolution measurements with well stablished modeling tools in São Paulo and Rio de Janeiro (see the works by Professors Paulo Artaxo, Maria Andrade, Luiz Pimentel, and others). There are at least 30 years of air pollution studies in Brazil and a "long road" of knowledge was built. This need to be valued. Your emission inventory does not consider many important features of the Brazilian fleet (content of ethanol in the gasoline, vehicular fleet aging, secondary roads, among others). You are applying different models that do not interact with each other. Your modeling results are far away from the observed concentrations. So, it is exaggerating to consider it as a pioneer study.

Page 5, lines 18-19. The authors said that "Emission factors for bi-articulated buses were extrapolated using information on fuel consumption provided by the Curitiba municipality". Please, clarify how was it possible to separate fuel consumption from one type of bus to another? Are bi-articulated buses using an exclusive type of diesel that allows that estimation? Page 5, lines 20-21. The authors said that "The use of biofuel lowers the PM and BC emissions by 50%, according to the U.S. Department of Energy (2018)". What is the relevance of this information for this study? Was this reduction

considered in your emissions? This is not clear.

Page 5, 30 – 34. What was the procedure to adjust EEA emission factors to the Brazilian reality? There is a recent work from Ibarra-Espinosa et al (2018) that could be consider in your work, since it applies very detailed procedure on building an emission inventory adjusted to the Brazilian conditions. (Ibarra-Espinosa, S., Ynoue, R., O'Sullivan, S., Pebesma, E., Andrade, M. D. F., and Osses, M.: VEIN v0.2.2: an R package for bottom–up vehicular emissions inventories, Geosci. Model Dev., 11, 2209-2229, https://doi.org/10.5194/gmd-11-2209-2018, 2018.)

Page 7, line 5. Actually, 10 x 10 km2 or 50 x 50 km2 refers to grid spacing, not resolution. The minimum resolution for these grids would be equivalent to 20 and 100 km2 (2 x delta x,y). The lowest grid spacing probably will represent the urban area of Curitiba by one or two grid points. In that situation, emissions will be poorly represented. Please comment on that matter.

Page 7, line 24. Please, comment on the constraints regarding the use of neutral stability in your simulations with OSPM.

Page 9, lines 3-9. The explanation for high BC concentrations during the weekends of 6-7 and 13-14 August is not clear. If the wind speed and emissions are low, how to explain the high levels of BC if you are ruling out this contributions?

Pages 9 and 10, section 3.3 and 3.4. These sections are very poorly explored. The authors give an impression of a very simple process to simulate. Results of the simulations are not in good agreement with the observed concentrations and the authors use linear correction rates to adjust the concentrations, instead of exploring the errors on their emissions. The processes involved are not linear. There are reactions involved that will be dependent on concentrations and environmental conditions. See for example the procedure described between lines 17 and 21.

Pages 13 and 14. As pointed out by the authors, many features were not included in

order to represent correctly the emissions of BC, PM2.5 and NOx. There are many arguments, but little basis for a correct guidance on procedures to be adopted for better public policies aimed at improving air quality. The observational part is rich, but the application of numerical models does not add important or useful information. Thus, the greatest recommendation regarding the article is that the observational part be used, but that the modeling part be withdrawn, since it gives the impression that the problem of air quality is of simple treatment, making adjustments here or there, neglecting important physical/chemical processes and replacing them with mere statistical procedures.

About the References

Missing references The references "World Medical Association, 2014" and "Zhang et al., 2015" are missing in the reference list.

Reference list The reference "Targino, A. C., Gibson, M. D., Krecl, P., Costa Rodrigues, M. V., dos Santos, M. M., & de Paula Corrêa, M. (2016). Hotspots of black carbon and PM2.5 in an urban area and relationships to traffic characteristics. Environmental Pollution, 218, 475-486. https://doi.org/10.1016/j.envpol.2016.07.027 ", was not cited in the text. Change the order of the references Wallace et al, 2011, and VISSIM, 2018.

---

## Author Comment (AC1) · 28 Jan 2019

Manuscript title (acp-2018-1094): Experimental and model assessment of PM2.5 and BC emissions and concentrations in a Brazilian city – the Curitiba case study Please find below the comments made by Reviewer 1. Authors' responses are given after each comment. The references to pages and lines are for the revised manuscript which includes the tracking changes.

Reviewer 1: "The manuscript carried out monitoring and model assessments in a Brazilian city Curitiba focusing on PM2.5 and Black Carbon emissions. It is an interesting and important research topic, however, there are information details lacking

in the current manuscript and it is like a report instead of a research paper. There is a large block of text (for example in the discussion and conclusion section) which is difficult for the readers to follow. I would suggest a major revision before we can consider for publication. It is suggested that the authors would improve the manuscript by adding sub‐headers at appropriate locations and also organize the tabulated data in form of bar/pie charts. Below are some specific comments:"

Authors' response: The authors acknowledge the constructive comments made by Reviewer 1. We have tried to improve the manuscript according to the specific suggestions, e.g. by adding sub-headers. Our responses to each of the specific comments are given here below, after each of the comments.

Introduction: 1. "Page 2 line 6: "One Specific SDG indicator, 11.6.2, for meeting this goal is the annual mean level of PM2.5 weighted by population. . ." This sentence seems incomplete. Are the authors trying to bring out that the annual mean level of population weighted PM2.5 reaches a certain value would meet the goal? Please consider re‐organize the sentence structure."

Authors' response: We have revised the sentence structure to assure it to be understandable, see page 2, lines 6 - 9.

Section 2.1 Study Area: 2. "Page 4 line 1: The authors mentioned there are four automatic stations within the municipality of Curitiba and four in the industrial area of the nearby city of Araucaria and an analysis of PM10 and NO2 from the official monitoring network was performed for three years from 2013 to 2015. The analysis include the data from the four automatic stations within the municipality of Curitiba but does it consider the data from the four stations in the industrial area of the nearby city of Araucaria? If not, what is the reason for excluding them in the analysis? Also please clarify whether there are missing data?"

Authors' response: The reason why we only discuss the four monitor stations inside the Curitiba municipality was because our assessment had to be limited to his area. Detailed traffic information was available for the Curitiba municipality, this is why our high-resolution model was restricted to the Curitiba municipality, not covering the neighboring Araucaria industrial area. We have stressed this areal limitation of our assessment in the end of Section 2.1 (page 4, line 5), and also at various locations in Section 2.2. The reviewer 1 also asks for a clarification if there are missing data. As for missing data at the four stations inside the Curitiba municipality, there is a column in Table 4 showing "Data capture" during the 2013-2015 period used as reference.

Section 2.2 Study Design: 3. "Page 4 line 7: Please be specific measured data from which stations are validating the emission inventory?"

Authors' response: In the last paragraph of Section 2.2 we have more clearly stated that the validation of the emission inventory was principally made by comparing street canyon increment at one station in the city center with the modeled impact of local traffic passing the street canyon, this leading to corrected emission factors for road vehicles. We then compared the urban background measurement data in the city center and in a residential area with high-resolution model output (based on the corrected emission factors for road vehicles) together with the long-range impact as determined by the regional model, revealing – as presented in the Result sections - reasonable results for BC but inconsistent results for PM2.5.

Section 2.3 Emission Inventory: "The emission inventory are developed mainly based on two economic sectors: industries and on‐road transport." 4. "Page 5 lines 5‐8: Detailed data for industrial sources are lacking. Please summarize and supplement."

Authors' response: We have added a description of the IAP industrial inventory, the type of industries found inside Curitiba and in the Araucária area just southwest of the city (page 5, lines 25-31). A summarized information on industrial PM emissions are found in the result Section 3.5 (page 12, line 14).

5. "It is observed that data has been provided in Tables 1‐3 for both public and

private vehicles. Please supplement bar charts to these tables for better illustration. The data on public transportation are quite detailed but for private vehicles, it is recommended to report the total number/types of vehicles and average speed as now seem there are only emission factors for private vehicles as in Table 3. What about uncertainty analysis?"

Authors' response: We made attempts to produce readable bar charts of Table 1-3, but it was found difficult to obtain the same amount of information presented in a pedagogic way and within a few diagrams. For example the following diagram only covers one of six data columns in Table 2 and we could not find a way of displaying all information in a single or a few bar charts.Moreover, those interested in the modeling part will find the absolute numbers of the emission factors to be useful. We thus believe that showing the tables makes this publication more informative.

We did not perform any uncertainty analysis of emission factors for private vehicles. Due to the much higher emission factors for diesel fueled LDVs and HDVs as compared to gasoline cars (Table 3), it is the percentages of those former type of vehicles that are critical (comment added page 6, line 31-32). As trucks are restricted to operate in the city center, we were suggested by traffic experts in Curitiba to work with 5% of diesel LDV and only 2% of HDV, this for most of the streets inside the ring road (only some transit roads excluded) and in particular for the street where we compared the street canyon model output with measurements. Our street canyon model evaluation resulted in a correction factor for private vehicle emissions, compensating for errors in vehicle fleet composition at this particular street as well as the emission factors taken from the literature. However, at other streets any different fleet composition from what we have assumed, will contribute to an erroneous emission and impact. We are clear in the manuscript that the lack of exact fleet composition for the private vehicles is a weak point in our assessment, e.g., in the Section 5 Limitations, page 18, lines 4-5.

6. "Page 5 line 23: Please describe more details about the VISSIM model and the corresponding input settings and data required for this model."

Authors' response: More details of the VISUM traffic model has been added on page 6, line 13-19.

7. "Page 5 line 25: Please provide information on the daily profiles adopted."

Authors' response: A new Fig. 2 has been included, revealing the daily profiles adopted.

Section 2.5 Dispersion Modelling: 8. "The EDGAR‐HTAP information include BC and OC emissions from what kind of sources, does the information include those BC release from vehicles and industries or wood burning within state of Parana?"

Authors' response: The EDGAR-HTAP should include those sources, but with a very coarse spatial resolution. We lack information to evaluate how complete the EDGAR-HTAP emissions of BC and OC are for the state of Paraná. We have added some details of the database in Section 2.5, page 8, lines 15-17. We conclude, after comparing the simulated and measure BC levels inside Curitiba, that the long-range BC contribution seems to be underestimated, see Section 4.7, page 16, line 13-16.

9. "It is mentioned that Gaussian Dispersion model incorporates a diagnostic wind model that takes into account surface roughness and building heights. As the building morphologies of the investigated city, are quite complex revealed in Fig 2 (right). Do the authors require inputting surface roughness and building height information or construct the CAD city model for OSPM model? If so, what surface roughness profiles/values and assumptions do the authors made to input in the OSPM model? Please include in the manuscript."

Authors' response: An explanation has been added of how surface roughness was calculated (page 8, line 23-24). The OSPM model uses wind speed at roof level, given by the diagnostic wind model, as input (added text page 9, line 4-5).

10. "Would the justification of choice of BRAMS for regional scale modelling (P.6 line 27‐29) be provided? How would be its accuracy compare with other major alternatives e.g. WRF?"

Authors' response: BRAMS is used together with CCATT model in an operation setup (http://meioambiente.cptec.inpe.br/index.php?lang=en) on 50x50 km2 grid resolution. It was natural to use the same meteorological model for the nesting down to 10x10 km2 over the Paraná state. It is not likely that using WRF with the same coarse grid resolution (10x10 km2) would have implied any substantial improvement in the CCATT model output, especially if one considers the important shortcomings in how CCATT was applied (no secondary PM, lack of detailed industrial BC emissions in the inventory etc). We thus argue that limitations in the regional CCATT model output are more found in the emission inventory than in the meteorological forcing. In addition, the CCATT-BRAMS modeling system, currently in the BRAMS 5.3 version, has more than 20 years of development in Brazil, with refinements and improvements allowing fast computations on multi-processer computers and parameterizations focused on the physical processes of South America.

Section 3 Results: 11. "Page 7 Line 28: There were no PM10 data reported from station CIC during the campaign period? Does it mean there is no PM10 release or there are other reasons behind? How about NO2 data at this station?"

Authors' response: There were technical problems for the PM10 monitor at the CIC station during the monitoring campaign, therefor no data were available. However, NO2 levels at CIC stations during the winter 2016 monitoring campaign were lower – 76% – as compared to the mean for the corresponding winter period (August) in 2013, 2014 and 2015. We have added this NO2 ratio at CIC station located in the more industrial area and close to the ring road, see Section 3.1, page 9, line 23.

12. "Page 7 Line 32: Would the authors please provide more information on the variation in long range transported pollution arriving to Curitiba."

Authors' response: We have introduced in Section 2.1 Study area a comment and a reference on possible long range contributions from the seasonal biomass burning in

northern and central Brazil (page 4, lines 7-10).

13. "Page 8 Line 21: Please describe what kind of technical failure caused the acquired data during fixed monitoring cannot be used."

Authors' response: There was an electronic problem with the pump with the consequence that it was not possible to keep a steady air flow. We could see clear signs of an interrupted air flow in the filter data, with days of data completely lost and with most of the remaining days a lower accumulated PM2.5 mass at the street level (close to traffic) as compared to at roof level. Due to this circumstance, that we were not sure about the accuracy of the data obtained, we preferred to discard these data.

14. "Section 3.4: It is mentioned that the street canyon dispersion model OSPM produced much smaller magnitude of PM2.5 and BC compared to measurement and regression analysis has been carried out to obtain correction factors. Please provide some brief information on the way that regression analysis were carried out. Also are the correction factors applicable to other places in Brazil? Have the authors tried other dispersion models such as AERMOD, ADMS or CFD approach, which might improve the magnitude prediction?"

Authors' response: A sentence has been added on the regression analysis (page 11, lines 32 – page 12, line 1-2). The conclusion from this Curitiba street canyon assessment was that the detailed information on public transport gave, together with emission factors from Europe, fairly accurate emissions. Similar information on public transport should be available in other Brazilian cities (a sentence added in Discussion page 14, line 30-31). However, for the private transport, with uncertainties both in vehicle fleet composition and vehicle technology, there is a risk for large errors and an attempt to determine site specific emission factors can be, like in Curitiba, necessary (sentence added in Discussion Section 4.5, page 15, lines 9-11). No other microenvironment model than OSPM have been used to simulate the dispersion of local traffic emissions inside street canyons in Curitiba. Since the PM2.5 and BC emission factors were determined from the assumption that NOx was correctly simulated by the model, it is not likely that the use of different models would have given other results (we have made linear corrections, see Authors' response to next comment).

15. "Section 3.4: Can the change of correction factor for BC for private vehicle emission by a factor of 5 be justified and conclude that it is not contributing by other factors like wrong fleet composition and modeling error? Is the new emission factor checked using an independent dataset of concentration measurement?"

Authors' response: In the assessment of local emission factors, our main assumption was that we found simulated NOx contributions to be similar to measured increments. This can be a coincidence, e.g. if dispersion/ventilation is underestimated and the emission factors overestimated. However, since NOx emission factors are more tested and more robust in the literature (as compared to PM and BC emission factors), we assumed that the similar results for measured and simulated contributions to NOx indicate that all steps in the simulation - including vehicle fleet composition, emission factors, dilution - were OK. If this is the case, then the large difference between simulated and measured BC contributions can't be explained by neither fleet composition nor model errors; instead the difference should have been created by erroneous emission factors. Comparison with independent dataset: Yes, while using the corrected emission factors estimated from the street canyon measurement and model simulations with OSPM, we find reasonable simulated BC contributions from local sources inside Curitiba also when comparing with urban background levels taken from independent stations (MD roof station, SC residential area). The local contribution is reasonable under the assumption that the regional model output of BC is underestimated, which seems possible when we only have the global EDGAR-HTAP database as input to BC emissions. Curitiba's own contribution to urban background BC levels seems to be about half of total measured BC, leaving space for a spatially homogeneous long-range contribution of approximately the same size as the local contribution ($\sim$1 $\mu$g m-3). Note that an earlier measurement campaign at the University campus showed a BC mean level just

above 2 $\mu$g m-3 (Polezer et al., 2018) i.e. the same urban background concentration level as for the two stations MD roof and SC of the present campaign.

Section 4 Discussions and conclusions: 16. "This section needs to be better organized around the main conclusions and highlights. At the moment, there is a large block of text with many information which is hard for the readers to follow. Please add sub‐headers at appropriate locations to break down the text and also present the important data in form of bar/pie charts instead of tables only. Please also list out important conclusions point by point and include a section on limitations of the study."

Authors' response: Nine sub-headers have been added, giving a structure of the Discussion and conclusion section more easy to follow. One text paragraph (concerning the mobile data collected with bikes) has also been moved to fit with this structure. The last sub-header is a point-by-point conclusion list. A new section 5 Limitations has been added.

17. "Please add a sub‐section that focus on discussion of comparison between model simulation results and measurement/monitoring data."

Authors' response: The comparison of simulated PM2.5 and BC levels with those measured in the two fixed urban background stations (MD roof and SC) is now part of a new sub-section named 4.7 Spatial distribution of PM2.5 and BC over Curitiba (comparison between simulated and measured concentration levels), page 15, line 1.

18. "EDGAR‐HTAP gives only 0.1deg x 0.1deg resolution. Do you mean local industrial BC emission is low when compared to other BC sources in Curitiba by providing a simulated BC urban background (P.10 line 32‐33)? Please justify?"

Authors' response: Industrial sources to BC emissions in the area just southwest of Curitiba are very coarsely described in the EDGAR-HTAP inventory and they constitute the only input of BC emission to the regional model. We also find the output over Curitiba to be very low, with BC contributions from sources outside the

Curitiba municipality of 0.06-0.07 $\mu$g m-3 in average, see Table 8. Local traffic sources contribute, according to the urban model, with a bit more than 1 $\mu$g m-3 of BC, however monitored mean levels are found above 2 $\mu$g m-3 both in the city center and in the residential area situated in the outskirts of the city. It is reasonable to think that the limited and coarse input of BC emissions from sources outside Curitiba can explain an underestimated long-range BC contribution. If local sources were behind the underestimated BC levels in the urban background, it would have been reasonable to see more varying levels at different locations (like we saw for PM2.5).

Please also note the supplement to this comment:
https://www.atmos-chem-phys-discuss.net/acp-2018-1094/acp-2018-1094-AC1-supplement.pdf

**Supplement:**

**Experimental and model assessment of PM$_{2.5}$ and BC emissions and concentrations in a Brazilian city – the Curitiba case study**

[revised manuscript text omitted]

The air quality in Curitiba is affected by emissions generated inside the city and by sources located outside in the Paraná state and in other parts the continent. An observational and modelling study conducted by Rosário et al. (2013) showed that smoke plumes generated during the biomass burning season in northern and central Brazil (August–October with and peak activity in September) reached the southern states, including the Curitiba region.

**2.2 Study design**

A combination of measurements and dispersion modeling was used to assess and map $PM_{2.5}$ and BC concentrations in Curitiba. The modeling had two purposes: validating the emission inventory through comparisons with measured data at a few stations, and obtaining a spatial distribution of the $PM_{2.5}$ and BC concentrations over the city. The comparison of monitored data and model output can be used for microenvironments impacted by one dominating source to allow an *in situ* determination of the emission factors, in this case for vehicles driven in the city.

The full assessment included the following components (see map in Fig. 1):

1.  Analysing the $NO_2/NO_x$ and $PM_{10}$ concentrations collected between 2013 and 2015 at the four IAP official monitoring sites in Curitiba: PAR, BOQ, STC and CIC (the monitoring network also includes stations outside Curitiba, however they were not used for comparisons with model output since the high resolution modelling only covered the Curitiba municipality).

2.  Developing an emission inventory for the city and the state of Paraná.

3.  Performing field campaign at two fixed sites (Fig. 1 left) aimed at:

    a) monitoring of $NO_x$, $PM_{2.5}$ and BC concentrations within a street canyon (Marechal Deodoro, hereafter MD) in the city center at two levels above ground: street (height of 5 m, hereafter 'MD street') and rooftop (height of 70 m, hereafter 'MD roof').

    b) monitoring of $PM_{2.5}$, BC, EC (elemental carbon) and OC (organic carbon) concentrations in a residential area (Sítio Cercado, hereafter SC) located 13 km from the city center, and 750 m from the heavily trafficked road BR-376, which is part of Curitiba's ring road ('Contorno' as local designation).

4.  Performing a monitoring campaign with instruments on-board bicycles to measure $PM_{2.5}$ and BC concentrations along different types of roads in the city center (see Fig. 1, right).

5. Implementing dispersion models at the regional, urban and street canyon scales to support the interpretation of the monitored data inside the Curitiba municipality.

6. Consolidating the street canyon data and model output to obtain real-world emission factors for PM$_{2.5}$ and BC for road transport in Curitiba.

7. Using the regional and urban modeling, together with the monitored data, to conduct a source apportionment of PM$_{2.5}$ and BC levels in the Curitiba municipality.

The modeling in a microenvironment – a street canyon – had the purpose to determine the impact on air quality levels by one dominating source, to allow an *in situ* determination of the emission factors, in this case for vehicles circulating in the city center. The comparison between measurements and model output was also extended to the urban background of the Curitiba municipality, comparing model results with the measurements in the city center and in a residential area, this in order to evaluate the emission inventory covering the Curitiba municipality, i.e., the area for which high resolution modelling was performed.

**2.3 Emission inventory**

The emission inventory developed for Curitiba considered PM$_{2.5}$, BC and NO$_x$ for two major economic sectors: industries and on-road transport. An attempt was made to collect data on the use of wood or coal by restaurants. However, the database gathered was incomplete in space, impeding the inclusion of this source in the emission inventory. Neither was it possible to obtain data on the residential use of wood stoves for cooking or heating.; hHowever, municipal authorities informed that residential wood combustion should be minimal, at least in the city center.

Industrial emission values of NO$_x$, SO$_2$ and PM$_{10}$ from large industrial sources were compiled from the official regional inventory that covers the state of Paraná (IAP, 2013), while the Curitiba municipality provided data for the inner-city small scale industries. Because Since these industrial inventories only included PM$_{10}$ emissions, we assumed that 70% of the PM$_{10}$ emitted by the industries in and around Curitiba consisted of PM$_{2.5}$ (Erlich et al., 2007). Information on industrial BC emissions was not available. All industrial emissions were treated as point sources with emissions coming out of a stack with characteristics given by the IAP inventory. The IAP inventory is based on a policy in which the industries are supposed to monitor themselves their own emissions and communicate to the official authorities. The IAP inventory reveals a concentration of sources in the southwest area of Curitiba, and in the neighbor city Araucária. Within the central parts of Curitiba there are just a few point sources, due to the urban planning of the 1970's and the creation of the Curitiba's Industrial Site located in the southwest region, moving almost all industries in the center to this new area downwind of the city according to the predominating winds. The Araucária city includes one of the biggest industrial sites of Brazil with a state-owned oil refinery together with several steel, cellulose/pulp and chemical industries.

[revised manuscript text omitted]
 probably clearly underestimated by the model (see Section 4.7 for the discussion of measured versus simulated BC levels in the urban background of Curitiba).

**4.5 Contribution to BC levels from local traffic**

An important aim with the ParCur assessment in Curitiba was to determine the emission of $PM_{2.5}$ and BC from sources inside the city. For that purpose, the street canyon measurements were used to evaluate in situ the emission characteristics of the vehicle fleet in Curitiba. With emission factors taken from the literature as input to the dispersion model, there was good agreement between the model outputs and the measured NOx concentrations, but the BC increment within the street canyon was underestimated by a factor of 3 (Table 6). The OSPM model has been used and evaluated in many urban environments and an extensive comparison showed good results for $NO_x$ (Ketzel et al., 2012). If assumed that the model gives reasonable results, then the good resemblance between simulated and monitored $NO_x$ values indicates that the $NO_x$ emission factors should be fairly accurate, but also that the BC emission factors are erroneous. With this as motivation, new emission factors for BC were determined through regression analysis. An interesting fact is that the regression analysis confirmed the magnitude of the BC emission factors of the buses, whose fleet composition, age and equivalent Euro class we know in detail. This is a result that can be used in other Brazilian cities where the characteristics of the public bus fleet is equally known in its details. However, the regression pointed out a large underestimation (factor of 5) of the mixed private vehicle fleet emissions, for which the composition and technical status was much less known. A factor of 5 of difference in an

emission factor may be seen as a too large discrepancy. However, BC emission factors show a large scatter and both the original 4 mg veh$^{-1}$ km$^{-1}$ and the corrected 19 mg veh$^{-1}$ km$^{-1}$ for the mixed private vehicle fleet circulating through the MD street canyon fit into "real-world" emission factors reported in the literature. Through chasing individual vehicles, BC emission factors spanned from 10 to 32 mg BC km$^{-1}$ for gasoline cars (Ježek et al., 2015), while for heavy-duty diesel vehicles they went from 53 mg BC km$^{-1}$ (Park et al., 2011) to values 10-fold higher (Wang et al., 2012). An assessment of real-world BC emission factors performed in a street canyon in Stockholm (Krecl et al., 2017) for the year 2006 (when BC levels within the street canyon were similar to those reported for M. Deodoro in Curitiba) found a BC emission factor for gasoline cars of 11 mg veh$^{-1}$ km$^{-1}$, which is significantly higher than the 0.15 mg veh$^{-1}$ km$^{-1}$ used originally in the present study (Table 3) and can therefore justify the higher emission factor coming out of the regression. A lesson learnt for other Brazilian cities is thus the necessity to assess local BC emission factors for the private vehicle fleet, using similar experiments as in Curitiba.

[revised manuscript text omitted]

**4.9 Summary of most important results from the experimental campaign:**

- Local emission factors for BC were determined for on-road vehicles.
- BC emissions from local traffic contributed to about half of total levels in Curitiba, the remainder being predominantly caused by sources outside the Curitiba municipality. Typical urban background  BC concentrations were  around 2.0 µg m$^{-3}$, raising to > 5.0 µg m$^{-3}$ close  to central street canyons.
- Urban background PM$_{2.5}$ concentrations in central Curitiba were 7.0-8.0 µg m$^{-3}$, doubling inside street canyons. In a residential area, concentrations three- to  four-fold higher  were observed, most likely due to local biomass or waste combustion.

**5 Limitations**

Some important limitations of this PM$_{2.5}$ and BC assessment in Curitiba have been identified:

- Measurement campaign limited in time (one month) and space (two fixed stations and mobile measurements along four routes covering an area of 4.8 km²)): An attempt was made to relate the  campaign data to the wintertime conditions observed during the three preceding years, this in order to assess the representativity of our collected PM$_{2.5}$ and BC data. The two fixed sites were well selected, however there is a need for future monitoring campaigns at additional locations (e.g., in the more polluted areas along the ring- road and towards the industrial zones just southwest of the Curitiba municipality).

- Technical failures during the campaign contributed to increased uncertainties in the $PM_{2.5}$ assessment in the street canyon. Also here a future campaign could give more reliable information on emission factors for the Curitiba vehicle fleet.
- While the model input information for simulating the impact of public transport was very detailed and of high quality, there was a lack of information on fleet composition for the private vehicles circulating the road network.
- Lack of emission information on some local sources of potential importance (e.g. restaurants, residential wood combustion), as well as regional information on $PM_{2.5}$ and BC emissions from the industrial sector.

[revised manuscript text omitted]

PTV GROUP, home page of the PTV GROUP, provider of traffic model VISUM: http://vision-traffic.ptvgroup.com/en-us/products/ptv-visum/ (last accessed 23 January 2019), 2019.

Rosário, N. E., Longo, K. M., Freitas, S. R., Yamasoe, M. A., and Fonseca, R. M.: Modeling the South American regional smoke plume: aerosol optical depth variability and Surface shortwave flux perturbation, Atmospheric Chemistry and Physics, 13, 2923-2938. https://doi.org/10.5194/acp-13-2923-2013, 2013.

Santos, E.: Curitiba, Brazil: Pioneering in developing Bus Rapid Transit and urban planning solutions, LAP Lambert Academic Publishing, 2014.

Santos, D. A. M., Brito, J. F., Godoy, J. M., and Artaxo, P.: Ambient concentrations and insights on organic and elemental carbon dynamics in São Paulo, Brazil, Atmos. Environ., 144, 226-233, https://doi.org/10.1016/j.atmosenv.2016.08.081, 2016.

Targino, A. C., Gibson, M. D., Krecl, P., Costa Rodrigues, M. V. C., dos Santos, M. M., & de Paula Corrêa, M. P.: Hotspots of black carbon and PM2.5 in an urban area and relationships to traffic characteristics. Environmental Pollution, 218, 475-486. https://doi.org/10.1016/j.envpol.2016.07.027, 2016.

Targino, A.C., and Krecl, P.: Local and regional contributions to black carbon aerosols in a mid-sized city in southern Brazil. Aerosol Air Qual. Res., 16, 125-137, https://doi.org/10.4209/aaqr.2015.06.0388, 2016.

TransportPolicy.net: https://www.transportpolicy.net/standard/brazil-heavy-duty-emissions/ (last accessed 19 September 2018January 2019), 2018a.

TransportPolicy.net: https://www.transportpolicy.net/standard/brazil-light-duty-emissions/ (last accessed 19 September 2018January 2019), 2018b.

United Nations Statistics Division: Global indicator framework for the Sustainable Development Goals and targets of the 2030 Agenda for Sustainable Development, 21 pp, https://unstats.un.org/sdgs/indicators/Global%20Indicator%20Framework%20after%20refinement_Eng.pdf, 2018 (accessed 19 January September 18, 20198).

U. S. Department of Energy: Biodiesel vehicle emissions, https://www.afdc.energy.gov/vehicles/diesels_emissions.html (last last accessed 18 September 2018), 2018.

VISSIM: software provided by PTV Group, http://vision-traffic.ptvgroup.com/en-us/home/ (last accessed 2919 Juneanuary 20198), 2018.

Wallace, L.A., Wheeler, A.J., Kearney, J., van Ryswyk, K., You, H., Kulka, R.H., Rasmussen, P.E., Brook, J.R., and Xu, X: Validation of continuous particle monitors for personal, indoor, and outdoor exposures, J. Expo. Sci. Environ. Epidemiol., 21, 49-64, https://doi.org/10.1038/jes.2010.15, 2011.

Walko, R. L., Cotton, W. R., Harrington, J. L., and Meyers, M. P.: New RAMS cloud microphysics parameterization. Part I: The single-moment scheme, Atmos. Res., 38, 29–62, 1995.

Wang, X., Westerdahl, D., Hu, J., Wu, Y., Yin, H., Pan, X., and Zhang, K. M.: On-road diesel vehicle emission factors for nitrogen oxides and black carbon in two Chinese cities, Atmos. Environ., 46, 45–55, https://doi.org/10.1016/j.atmosenv.2011.10.033, 2012.

WHO: World Health Organization Review of Evidence on Health Aspects of Air Pollution—REVIHAAP Project, WHO Regional Office for Europe: Copenhagen, Denmark, available at : http://www.euro.who.int/__data/assets/pdf_file/0004/193108/REVIHAAP-Final-technical-report-final-version.pdf (last access accessed 1914 April January 20198), 2013.

WHO: Ambient air pollution: A global assessment of exposure and burden of disease, WHO Library Cataloguing-in-Publication Data, 121 pp, available at: http://www.who.int/phe/publications/air-pollution-global-assessment/en/ (last accessed 19 January4 April 2018 2019), 2016.

World Medical Association, WMA statement on the prevention of air pollution due to vehicle emissions. https://pdf-it.dev.acw.website/please-and-thank-you?url=https://www.wma.net/policies-post/wma-statement-on-the-prevention-of-air-pollution-due-to-vehicle-emissions/&pdfName=wma-statement-on-the-prevention-of-air-pollution-due-to-vehicle-emissions 2014 (last accessed 18 September 2018), 2014.

Zhang, Q., Qiu, Z., Chung, K.F.: Link between environmental air pollution and allergic asthma: East meets West. J. Thorac. Dis., 7, 14–22, https://doi.org/10.3978/j.issn.2072-1439.2014.12.07-, 2015.

**Fig. captions**

[Figure]

**Fig. 1:** Maps showing the location of: i) Left: the four official monitoring stations (CIC, PAR, BOQ, STC), the site in the residential area (SC), and the meteorological station (MET), ii) Right: the street canyon site (MD), and the four biking trajectories used for the mobile measurements.

[Figure]

**Fig. 2** Measured traffic volumes (veh h⁻¹) from the so called speed traps (speed radar data) at 230 locations in Curitiba. To the left raw data, to the right the aggregated time variations for different days of the week used in the dispersion modelling.

[revised manuscript text omitted]

---

## Author Comment (AC2) · 28 Jan 2019

Manuscript title (acp-2018-1094): Experimental and model assessment of PM2.5 and BC emissions and concentrations in a Brazilian city – the Curitiba case study

Please find below the comments made by Reviewer 2. Authors' responses are given after each comment. The references to pages and lines are for the revised manuscript which includes the tracking changes (submitted as Supplement).

Reviewer 2: General comments "The article presents important results regarding campaigns of measures in the scope of the ParCur project. The observational part was

made based on measures close to the surface, in fixed and "mobile" stations (bicycles), and at the level of 70 m above the ground. This is the richest and most important part of the article. Also, three different models were applied to try to represent the concentrations of Black Carbon, Particulate Material 2.5 and Oxides of Nitrogen. Despite three different approaches to modeling, little useful information has been obtained, making several approaches, both in terms of emissions and in terms of the results obtained, and is therefore the weak part of the article. The general recommendation is to keep only the observational part of the work, with the modeling part being excluded. Here are some specific points that led to this conclusion."

Authors' response: The authors acknowledge the constructive comments made by Reviewer 2. As for the general recommendation to keep only the observational part of the work, we feel that this comes as a consequence of our first manuscript version failing to clearly describe the methodology applied. We hope that our revised manuscript, together with our responses to the reviewer's specific comments here below, will convince the reviewer 2 of the meaningfulness of the procedures where our conclusions are based on a combination of measurement and model results.

Specific comments 1. "Page 2, line 19: correct the units "$\mu$g m-3"

Authors' response: Corrected.

2. "Page 2, line 29: Is the year of the reference correct? In the reference list it appears as 2014"

Authors' response: It should be 2014, corrected page 2, line 29 and line 30 (note also that the URL has been updated).

3. "Page 3, lines 17 – 20. The authors said that "As far as we know, this is a pioneering study in South America with the integration of fixed and mobile high spatio-temporal resolution PM2.5 and BC measurements, the development of emission inventories and the implementation of modeling tools at different spatial scales for validating the emissions and to determine the spatial distribution of pollutant concentrations across the city." Maybe you are referring to studies about Black Carbon over Curitiba, but, in fact, there are at least a dozen of studies combining high resolution measurements with well stablished modeling tools in São Paulo and Rio de Janeiro (see the works by Professors Paulo Artaxo, Maria Andrade, Luiz Pimentel, and others). There are at least 30 years of air pollution studies in Brazil and a "long road" of knowledge was built. This need to be valued. Your emission inventory does not consider many important features of the Brazilian fleet (content of ethanol in the gasoline, vehicular fleet aging, secondary roads, among others). You are applying different models that do not interact with each other. Your modeling results are far away from the observed concentrations. So, it is exaggerating to consider it as a pioneer study."

Authors' response: We agree that this formulation is provocative and we have replaced the word "pioneering" with "innovative". We still claim that the way existing activity data, measurements and models have been used together is new and has, to our knowledge, not been published for Brazilian cities before. However, we do not say that monitoring and modeling in general is new in Brazil, we do reference a lot of Brazilian studies and the reviewer 2 is correct in her/his list of important contributions, especially from São Paulo and Rio de Janeiro. What we claim:

a) High spatio-temporal resolution PM2.5 and BC refer to the biking experiment, whose application in Brazil has been developed by co-authors of this manuscript.

b) Air quality modeling in Brazil: According to an overview given by Andrade et al. (Air quality in the megacity of São Paulo: Evolution over the last 30 years and future perspectives) there is a history of 20 years of air quality modeling, first with offline models and later with mesoscale inline/online models of the type used in this study (BRAMS-CCATT) and WRF-Chem. However, we have not found high spatial resolution model studies of the type used in the present study, neither have we seen street canyon modeling used to assess the traffic impact on the micro-scale.

[Figure]

c) We agree that our emission inventory is incomplete, covering traffic and industry for PM but only traffic for BC. However, traffic emissions are locally determined by the street canyon modeling and monitoring (making unnecessary knowing the aging of the fleet and specific fuel composition). Larger secondary roads are covered by the VISUM traffic model (see added details on this input data on page 6, lines 5-13) for private transport and the detailed information of the public transport emissions is on a unique level covering all streets where public transport is operating.

d) The high spatial resolution modeling shows PM2.5 contributions of local sources that fit the measured urban background in the city center, while the measurements in the residential area clearly point out to an unidentified local source of PM2.5. The main purpose with the modeling in this assessment is to explain the role of the local and regional contributions from sources that are included in the inventory. Any differences between model output and measured total levels are discussed in terms of potential other sources and deficiencies in the different inventories. We claim that this information is more easily and less costly obtained by using dispersion models together with measurements.

4. "Page 5, lines 18-19. The authors said that "Emission factors for bi-articulated buses were extrapolated using information on fuel consumption provided by the Curitiba municipality". Please, clarify how was it possible to separate fuel consumption from one type of bus to another? Are bi-articulated buses using an exclusive type of diesel that allows that estimation? Page 5, lines 20-21. The authors said that "The use of biofuel lowers the PM and BC emissions by 50%, according to the U.S. Department of Energy (2018)". What is the relevance of this information for this study? Was this reduction considered in your emissions? This is not clear."

Authors' response: Local authorities (URBS) reported type of fuel and fuel consumption for nine types of buses, bi-articulated being one of them using either diesel (S10) or biodiesel (B100). Discussions with emission specialists for the Stockholm bus fleet showed that they were of the opinion that the 50% reduction of PM and BC emissions

while using 100% biofuel, as indicated by the US reference, was reasonable. Yes, the emission factors were reduced in this way for the buses using biofuel. We have changed the text to make this more clear (page 6, lines 10-11).

5. "Page 5, 30 – 34. What was the procedure to adjust EEA emission factors to the Brazilian reality? There is a recent work from Ibarra-Espinosa et al (2018) that could be consider in your work, since it applies very detailed procedure on building an emission inventory adjusted to the Brazilian conditions. (Ibarra-Espinosa, S., Ynoue, R., O'Sullivan, S., Pebesma, E., Andrade, M. D. F., and Osses, M.: VEIN v0.2.2: an R package for bottom–up vehicular emissions inventories, Geosci. Model Dev., 11, 2209-2229, https://doi.org/10.5194/gmd-11-2209-2018, 2018.)"

Authors' response: We thank the reviewer for giving this interesting and recent reference. The bottom-up approach and the emission modeling in VEIN is very similar to the mobile emission inventory (part of software package Airviro) used in our study. In our study, activity data for private vehicles come from a traffic model, as suggested in VEIN. For public transport we have simulated each bus line using its time table, which should give a more precise information on vehicles per hour passing a certain road link. VEIN suggests the use of COPERT emission factors, using vehicle speed. There is in Europe a debate of using vehicle speed or signed speed limit together with the traffic situation (type of flow, if free, congested, saturated with stop-and-go etc) as a parameter to modify the emission factors, an approach developed in the ARTEMIS project and implemented in the HBEFA emission factors. For buses we used emission factors that considered traffic situations (we assumed saturated conditions), however the EEA emission factors used for private fleet did neither use speed nor traffic situation as a parameter. The main difference in our approach in Curitiba, as compared to VEIN, is that we, after simple assumptions on vehicle technology and speed, confirmed the emission factors through the street canyon modeling and monitoring experiment.

6. Page 7, line 5. Actually, 10 x 10 km2 or 50 x 50 km2 refers to grid spacing, not resolution. The minimum resolution for these grids would be equivalent to 20 and

100 km2 (2 x delta x,y). The lowest grid spacing probably will represent the urban area of Curitiba by one or two grid points. In that situation, emissions will be poorly represented. Please comment on that matter.

Authors' response: The resolution of a meteorological model describing weather events is various time larger than the grid resolution. We have changed the wording describing the resolution of the regional model (page 8, line 13-14), from "spatial resolution" to "grid resolution". However, the regional model was not used to map the air quality over Curitiba, only to provide the long-range transport. The local model had a grid resolution of 200x200 m, while emissions grids were generated with a higher resolution (100x100 m). The Gaussian model gives a continuous decay away from each source, although evaluated only at the 200x200 m grid points. Thus it is fair to say that the horizontal resolution over the city is at least equal and most likely better than the 200x200 m of the evaluation grid.

7. "Page 7, line 24. Please, comment on the constraints regarding the use of neutral stability in your simulations with OSPM."

Authors' response: The diagnostic wind model used to generate the wind at roof level takes stability into account. However within the street canyon neutral conditions are assumed, this being part of the OSPM assumptions. To our knowledge this is not an important constraint since turbulence in a street canyon is mainly mechanically-generated. During strong cooling (working towards stable conditions) the heat collected inside the street canyon, together with the turbulence generated by the traffic and the street canyon vortex itself should counteract any stratification tendencies within the street canyon. There has been for street canyons located at very high latitudes (northern Norway) a case indicating a smaller deviation from measurements that could be related to the neutral assumption instead of stable conditions, but this has been shown not to constitute a problem while using OSPM in more mid-latitude cities like Copenhagen. As for convective conditions, it is likely to expect in tropical cities some effects, but there is to our knowledge nothing published around this effect on OSPM results.

[Figure]

It is difficult to think that the convective effects should invalidate OSPM simulations of wintertime Curitiba street canyon concentrations.

8. "Page 9, lines 3-9. The explanation for high BC concentrations during the weekends of 6-7 and 13-14 August is not clear. If the wind speed and emissions are low, how to explain the high levels of BC if you are ruling out this contributions?

Authors' response: These high levels occur during weekend nights when traffic emissions are low. Even if the peaks in the city center (MD) and in the residential area (SC) come at the same nights, they are not following each other exactly in time. We interpret this as an effect of one or various unidentified source(s), inside or just outside the city (if it was far away the responses at the two stations should have followed a more similar time variation).

9. "Pages 9 and 10, section 3.3 and 3.4. These sections are very poorly explored. The authors give an impression of a very simple process to simulate. Results of the simulations are not in good agreement with the observed concentrations and the authors use linear correction rates to adjust the concentrations, instead of exploring the errors on their emissions. The processes involved are not linear. There are reactions involved that will be dependent on concentrations and environmental conditions. See for example the procedure described between lines 17 and 21."

Authors' response: The regional model output concentrations, as given in Section 3.3, have not been corrected, only discussed in terms of uncertainties in the emission inventory outside the city. Based on the result section 3.3, discussing Table 5, it is premature to conclude on an underestimation of the long-range BC contribution. We have edited the text in Section 3.3 to only indicate large uncertainties due to the use of the coarse EDGAR-HTAP emission data (page 11, lin 14-16). The conclusion that those emissions are likely to be underestimated comes in connection to the comparison between measured and simulated urban background BC levels, i.e. in Section 4.7, page 16, lines 13-15. As for Section 3.4, which describes the comparison of measured

and simulated contributions of local traffic inside a street canyon, we claim that NOx, PM2.5 and BC contributions from the local traffic can be assumed inert as long as they stay inside the street canyon. Consequently the assumption of a linear relationship between emissions and concentrations in the dispersion model output can be justified.

10. "Pages 13 and 14. As pointed out by the authors, many features were not included in order to represent correctly the emissions of BC, PM2.5 and NOx. There are many arguments, but little basis for a correct guidance on procedures to be adopted for better public policies aimed at improving air quality. The observational part is rich, but the application of numerical models does not add important or useful information. Thus, the greatest recommendation regarding the article is that the observational part be used, but that the modeling part be withdrawn, since it gives the impression that the problem of air quality is of simple treatment, making adjustments here or there, neglecting important physical/chemical processes and replacing them with mere statistical procedures."

Authors' response: Following our earlier answers, we claim that the integrated analysis with both modeling and monitoring exercises give important information on sources and spatial distributions of PM and BC concentrations. In the text we admit the major shortcomings of the models, the emission inventories and the monitored data; also lifting forward some unique features like the very detailed emission inventory for public transport. Our intention has been to show the contributions from long-range (sources outside the city) and local sources inside the city. We have been able to explain a reasonable local contribution to BC, which together with an underestimated and spatially homogeneous long-range contribution fit to measured BC levels in the urban background. For PM2.5 the same comparison points out unidentified sources contributing to locally raised PM2.5, not possible to explain with a spatially uniform long-range contribution. Our experience is that for fairly inert urban pollutants like NOx, BC and - for smaller cities - PM, the large uncertainty is found in the emission inventories, not in dispersion model formulations.

About the References 11. "The references "World Medical Association, 2014" and "Zhang et al., 2015" are missing in the reference list."

Authors' response: In our version of the submitted manuscript these two final references are found on page 20. We will assure that they are visible in the revised manuscript to be uploaded.

12. "The reference "Targino, A. C., Gibson, M. D., Krecl, P., Costa Rodrigues, M. V., dos Santos, M. M., & de Paula Corrêa, M. (2016). Hotspots of black carbon and PM2.5 in an urban area and relationships to traffic characteristics. Environmental Pollution, 218, 475-486. https://doi.org/10.1016/j.envpol.2016.07.027 ", was not cited in the text. "

Authors' response: Correct, the quoting of this reference had by mistake been eliminated. We have introduced it again, first on page 3, line 2; secondly on page 7, line 29.

13. "Change the order of the references Wallace et al, 2011, and VISSIM, 2018."

Authors' response: The reference "VISSIM, 2018" has been renamed and moved to "PTV GROUP, 2019".

Please also note the supplement to this comment:
https://www.atmos-chem-phys-discuss.net/acp-2018-1094/acp-2018-1094-AC2-supplement.pdf